# Language Models can Solve Computer Tasks

**Geunwoo Kim**
University of California, Irvine
`kgw@uci.edu`

**Pierre Baldi**
University of California, Irvine
`pfbaldi@ics.uci.edu`

**Stephen McAleer**[*]
Carnegie Mellon University
`smcaleer@cs.cmu.edu`

## Abstract

Agents capable of carrying out general tasks on a computer can improve efficiency and productivity by automating repetitive tasks and assisting in complex problem-solving. Ideally, such agents should be able to solve new computer tasks presented to them through natural language commands. However, previous approaches to this problem require large amounts of expert demonstrations and task-specific reward functions, both of which are impractical for new tasks. In this work, we show that a pre-trained large language model (LLM) agent can execute computer tasks guided by natural language using a simple prompting scheme where the agent **R**ecursively **C**riticizes and **I**mproves its output (RCI). The RCI approach significantly outperforms existing LLM methods for automating computer tasks and surpasses supervised learning (SL) and reinforcement learning (RL) approaches on the MiniWoB++ benchmark. We compare multiple LLMs and find that RCI with the InstructGPT-3+RLHF LLM is state-of-the-art on MiniWoB++, using only a handful of demonstrations per task rather than tens of thousands, and without a task-specific reward function. Furthermore, we demonstrate RCI prompting's effectiveness in enhancing LLMs' reasoning abilities on a suite of natural language reasoning tasks, outperforming chain of thought (CoT) prompting with external feedback. We find that RCI combined with CoT performs better than either separately. Our code can be found here: `https://github.com/posgnu/rci-agent`.

## 1 Introduction

A long-standing goal in artificial intelligence has been to create generally-intelligent agents that can accomplish cognitive tasks as well as humans. Such agents should be able to solve any computer task a human can by communicating via natural language. By automating repetitive tasks and providing assistance in complex problem-solving, generally-intelligent virtual agents may radically increase productivity.

Recently, large language models (LLMs) have shown remarkable in-context learning capabilities across a variety of domains and tasks [12, 69, 5, 17, 26, 64, 8, 46, 6]. Although LLMs can impressively manipulate text and can use high-level API tools [59, 48, 41], previous approaches to using LLMs that directly take keyboard and mouse actions on computers have had difficulty compared to imitation learning and reinforcement learning approaches [24]. LLMs that take keyboard and mouse actions on computers face a number of obstacles, such as ensuring that generated actions are task-appropriate (task grounding), feasible in the agent's current state (state grounding), and admissible to be executed (agent grounding).

---

[*]Corresponding author.

37th Conference on Neural Information Processing Systems (NeurIPS 2023).

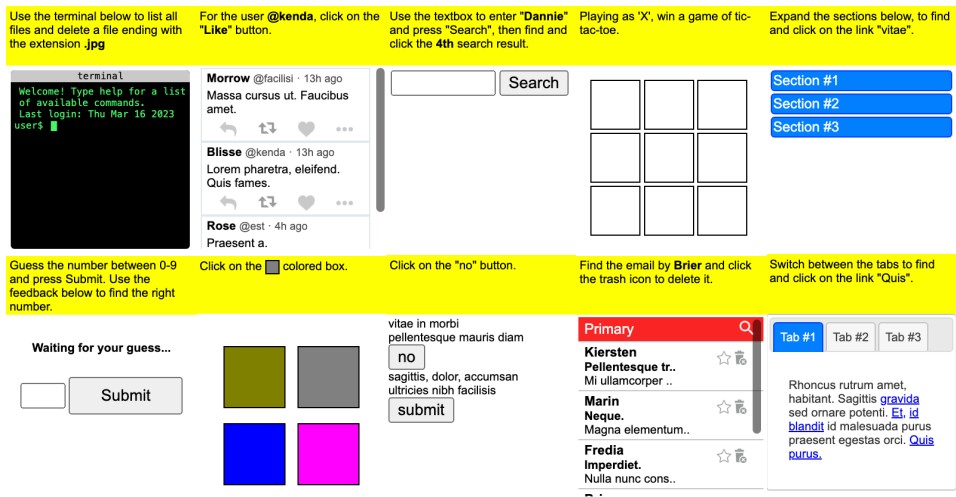

Figure 1: MiniWoB++ environment. Every task contains a natural language prompt in yellow. The agent then uses keyboard strokes and mouse clicks to accomplish the task.

The previous best-performing approaches for taking actions on computers have not used LLMs. Instead, they have trained networks from scratch to predict actions given prompts and screenshots or DOM information, either via supervised learning (SL) from expert demonstrations, reinforcement learning (RL) on a handcrafted reward signal, or both (SL+RL) [30]. Although SL+RL works well on a number of individual computer tasks, since it requires expert data and a reward function for every task, it has not been shown to generalize to novel tasks in a few-shot setting.

In this work, we show that a pre-trained LLM agent can successfully execute computer tasks guided by natural language. Our method employs a simple prompting scheme, which we call Recursive Criticism and Improvement (RCI), that significantly outperforms existing LLM methods for automating computer tasks. RCI works by first having the LLM generate an output based on zero-shot prompting. Then, RCI prompts the LLM to identify problems with the given output. After the LLM has identified problems with the output, RCI prompts the LLM to generate an updated output.

When applying RCI to computer tasks, we improve task grounding, state grounding, and agent grounding sequentially. Firstly, task grounding prompts the LLM with the task text, instructing it to generate a high-level plan. Secondly, state grounding connects high-level concepts derived from the task grounding step with actual HTML elements present in the current state, subsequently outputting the appropriate action. Finally, agent grounding ensures the correct formatting of the action output obtained from the state grounding step. RCI is applied to each of these three steps; however, we find that critiquing the state-grounding step is only necessary once.

We evaluate the RCI approach on the MiniWoB++ benchmark [61], and show it surpasses existing SL, RL, and LLM approaches. Furthermore, it proves itself to state-of-the-art compared to existing methods, using only a small number of demonstrations per task instead of tens of thousands, and without relying on a task-specific reward function. This significant reduction in required demonstrations and the elimination of task-specific reward functions make our method more practical and accessible for new tasks. Furthermore, as the capabilities of LLMs continue to improve, one can expect the performance of our method to improve as well.

In addition to its success in automating computer tasks, we also showcase the effectiveness of RCI prompting in enhancing the reasoning abilities of LLMs on a suite of natural language reasoning tasks. When external feedback is given, our method achieves a significant performance increase over zero-shot prompting and slightly improves upon chain-of-thought [73] (CoT) prompting. Interestingly, RCI and CoT have a synergistic effect, and their combination outperforms all other methods.

In summary, our work presents a new powerful and practical approach to enabling LLM agents to execute computer tasks guided by natural language. The RCI prompting scheme not only outperforms

Figure 2: Illustrative examples of explicit RCI prompting and baseline prompting approaches on the GSM8K dataset. RCI prompting effectively addresses logical errors that arise in the baseline prompting approaches. Prompts text is displayed in violet color.

previous methods in computer tasks, but also improves reasoning abilities for LLMs more broadly, making it a significant contribution in the development of intelligent agents.

## 2 Methods

### 2.1 RCI Prompting

The self-critiquing ability of LLMs has demonstrated that LLMs can find errors in their own output by themselves [58, 20, 3]. In light of this, we introduce a simple reasoning architecture called RCI prompting, where we prompt LLMs to find problems in their output and improve the output based on what they find. This architecture is designed to further enhance the reasoning ability of LLMs by inserting a critique step before generating the final answer. Figure 2 compares example traces of RCI prompting and baseline prompting methods on GSM8K dataset where language models should answer grade school math problems. While baselines elicit answers with a single step of prompting, RCI consists of two steps: criticize the previous answer (*e.g.,* "Review your previous answer and find problems with your answer") and improve the answer based on the critique (*e.g.,* "Based on the problems you found, improve your answer"). In this way, RCI prompting finds errors (*e.g.,* the overall sum of money only considered Valerie and her brother) in the previous answer and generates an improved answer (*e.g.,* money from Valerie's mother is included in the total) conditioned on the critique. The iterative process of RCI can be continued until specific conditions are satisfied, which could include receiving feedback from the environment, reaching the maximum predetermined number of iterations, or adhering to certain heuristics. We define two approaches for achieving RCI: explicit RCI and implicit RCI. Explicit RCI includes the critique in the prompt to generate improved output and implicit RCI updates the previous output directly without sampling a critique explicitly. Examples of explicit RCI and implicit RCI applied to computer tasks are illustrated in Figure 3 where the action plan is improved based on an explicit critique while actions are updated without an explicit critique. This process is described in more detail in the following section.

### 2.2 RCI for Computer Tasks

In this section we describe the application of RCI to computer tasks via a decomposition of action selection into three reasoning steps: task grounding, state grounding, and agent grounding. The first step, task grounding, involves generating a plan for task-solving and conditioning actions on this plan, with RCI being used to improve the plan's success rate. The state grounding subsection discusses the importance of grounding actions in the environment for language-based agents and how implicit RCI is used to refine task-grounded actions to be feasible in the current state. Lastly, the agent grounding

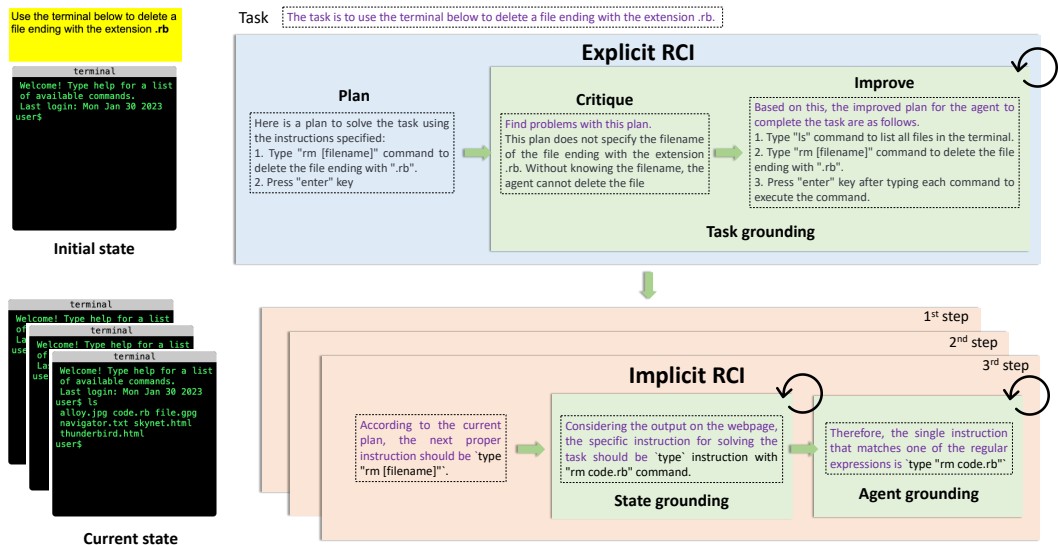

Figure 3: An illustrative execution trace of the agent for terminal tasks with RCI prompting. The language model generates a step-by-step plan for the high-level task described in natural language, which in this case involves using the terminal to delete a file ending with ".rb". We then run an explicit RCI on this plan, where we sample an improved plan based on the critique and the previous plan, resulting in an improvement in the task-grounding of the plan. For each step, we first sample the task-grounded action that follows the improved plan, and then the implicit RCI updates the task-grounded actions sequentially to provide state-grounding and agent-grounding. Finally, the agent-grounded action is executed by the instruction-following agent on the environment. The prompts are highlighted, and the remaining text shows the outputs generated by the language model.

step focuses on ensuring that actions are admissible for the computer agent by employing implicit RCI and conditioning agent-grounded actions on the current state, task, and other grounded actions, with a loop count set to optimize performance.

### 2.2.1 Problem Setting

We assume that we are given an instruction-following computer agent that can execute a set of admissible actions given some natural language instructions. An instruction that is not part of the admissible actions will be ignored. At every step, we receive a high-level natural language task prompt and a state of the environment. Given the current state and task, we sample the most probable action from LLMs. The generated natural language action is then fed into the computer agent. Sampling the actions in a fully generative manner presents a challenge, as the actions must consider the given task, feasibility in the current state, and admissibility for the computer agent simultaneously. Therefore, we propose decomposing this action sampling into three reasoning steps each of which considers task grounding, state grounding, and agent grounding. Task grounding improves actions to be more effective in solving the given task, state grounding ensures the feasibility of actions in the current state, and agent grounding considers the executability of actions given the specification of the computer agent. We first sample a step-by-step plan to solve the given task which improves the task grounding. Next, the task-grounded action is sampled conditioned on the current state, task, and the generated plan. The state-grounded actions is generated conditioned on the task-grounded action. If the task-grounded action is not executable by the computer agent, the agent-grounded action is sampled. For each sampling of grounded action, we use RCI prompting to make LLM consider some specific information for grounding.

### 2.2.2 Grounding Language Model in Computer Tasks

**Task grounding.** In the action sampling process, the first step involves generating a plan of actionable steps for task solving from LLMs. Subsequently, actions are sampled from the same LLMs, taking into account the present state, task, and generated plan. The benefits of conditioning on

the plan for improved grounding of actions are twofold. First, it enables LLMs to identify the stage of task solving at which the agent is located, serving as a memory module. Second, we can perform explicit RCI on the generated plan to further improve the plan's success rate. Although the number of explicit RCI loops can be arbitrary, we observe that a single pass of explicit RCI suffices for most of MiniWoB++ tasks.

**State grounding.**   In language-based agents, grounding actions in the environment is a crucial step to enable real-world task performance. The aim of this phase is to enhance the task-grounded actions to be feasible in the current state. Although the actions generated in the preceding phase may align with the task, they may lack the specificity required to be executed in the current context. For example, if the assigned task is to forward an email from Bob to Alice and the action obtained from the task grounding phase is to click on an email from Bob in the email inbox, it is necessary to establish a connection between the abstract concept of "email from Bob" and the concrete element, such as the email heading, in the current webpage state represented by HTML. To achieve this goal, we perform the implicit RCI and prompt the LLMs to consider the current state, which subsequently outputs refined state-grounded actions. Moreover, the state-grounded action is additionally conditioned on the task-grounded action. We avoid repeating the implicit RCI cycle more than once as it does not impact the success rate based on our observations.

**Agent grounding.**   To ensure the successful integration of language-based methodologies in decision-making processes, it is imperative to establish a scalable framework that guarantees the admissibility of actions derived from the language model. While the preceding steps of sampling produce a state-grounded action that is both feasible and grounded in the task, it may not be executable by the agent due to issues such as improper formatting. To address this, Implicit RCI is employed, whereby an agent-grounded action is sampled conditioned on the current state, task, task-grounded action, and state-grounded action. The LLMs are prompted to consider specifications of the computer agent. The implicit RCI is repeatedly run until the resulting action is executable, with a maximum loop count set to limit the number of iterations. Empirical analysis on MiniWoB++ tasks suggests that setting the loop count to 3 yields optimal performance.

## 3   Evaluation

### 3.1   Reasoning tasks

In our grounding enhancement process, RCI prompts the LLM to criticize its prior output, considering the given context (*e.g.,* current task, state, and agent), which ultimately leads to improved output. We first demonstrate the effectiveness of RCI prompts in augmenting the reasoning capabilities of LLMs across a range of reasoning benchmarks. We compare RCI to Chain-of-Thought (CoT) prompting, a state-of-the-art method recognized for its effectiveness in reasoning tasks.

Specifically, we compare our approach with Few-Shot-CoT [73] where a few chain-of-thought demonstrations are given as examples in prompting, and Zero-Shot-CoT [33] that elicit multiple reasoning steps by simply adding "Let's think step by step" to the prompt. Following Kojima et al. [33], our evaluation is conducted with 8 datasets from two categories of reasoning: arithmetic and commonsense. Please refer to Appendix C.2 for a comprehensive depiction of the datasets. We use the same experimental setting with their answer extraction method except that we use InstructGPT-3 + RLHF (*gpt-3.5-turbo*) as the underlying language model. We use the same prompts that CoT uses and we also use the answer cleansing approach used in CoT, but we only used answer extraction prompting in zero-shot CoT experiments. We also use the same few-shot examples that were introduced in [73] to evaluate Few-Shot CoT's performance on five arithmetic reasoning tasks. A threshold is established by setting the maximum number of RCI loops to two, terminating the loop once the output aligns with the ground-truth data. We observed that in the absence of this external feedback mechanism, the RCI process is prone to false negative critics, subsequently leading to a decrease in performance. Experimental results indicate that RCI without external feedback achieves zero-shot performance in half of the benchmark tests, but underperforms in others, as shown in Appendix 17.

**Comparison with Zero-Shot.**   RCI prompting is better at solving reasoning tasks compared to zero-shot prompting. Table 1 summarizes the accuracy of our approach (Zero-Shot + RCI) and standard zero-shot prompting for each reasoning benchmark. Zero-Shot + RCI substantially outperforms the

standard prompting in all benchmarks including arithmetic (GSM8K, MultiArith, AddSub, AQUA, SVAMP, SingleEq) and common sense (CommonSenseQA, StrategyQA) tasks. RCI prompting even achieves score gains from two arithmetic reasoning tasks (SingleEq and AddSub), which do not require multi-step reasoning. This distinguishes our RCI prompting from the previous CoT prompting methods [73, 33] that are not useful in simple reasoning tasks. It is also worth noting that RCI prompting achieves a significant performance gain in commonsense reasoning tasks (CommonSenseQA and StrategyQA). While Wei et al. [73] reported that only a substantially large PaLM (540B) model can benefit from Few-Shot-CoT, RCI prompting can provide performance gain even with a smaller InstructGPT-3 + RLHF (175B) model.

| | Arithmetic | | | | | | Common Sense | |
|---|---|---|---|---|---|---|---|---|
| | GSM8K | MultiArith | AddSub | SVAMP | SingleEq | AQuA | CommonSenseQA | StrategyQA |
| Zero-Shot | 77.95 | 94.48 | 88.58 | 80.70 | 86.61 | 60.23 | 64.56 | 48.81 |
| Zero-Shot + RCI | **85.43** | **97.64** | **89.76** | **84.65** | **94.49** | **67.32** | **68.11** | **61.81** |

Table 1: RCI prompting increases the reasoning capability of LLMs on all of eight reasoning benchmarks.

**Comparison with Chain-of-Thought.** The performance results of RCI and CoT baselines on arithmetic reasoning tasks are summarized in Table 2. Notably, Zero-Shot + RCI outperforms Zero-Shot CoT and Few-Shot CoT without any CoT prompting in four tasks except *MultiArith*. In *MultiArith* tasks, where most of the standard prompting's answers are correct (96.06%), RCI prompting does not yield significant performance gains. RCI prompting has a synergistic collaborative impact on the two CoT baselines. Namely, Zero-Shot CoT + RCI and Few-Shot CoT + RCI attain the highest scores on four out of the five tasks. These findings suggest a promising avenue for future research: combining RCI with other prompting methods for CoT, such as self-consistency [58].

| | GSM8K | MultiArith | AddSub | SVAMP | SingleEq |
|---|---|---|---|---|---|
| Zero-Shot | 78.35 | 96.06 | 85.83 | 78.35 | 91.34 |
| Zero-Shot + RCI | 85.43 | 97.64 | 89.76 | 84.65 | **94.49** |
| Zero-Shot CoT | 82.28 | 96.85 | 83.86 | 79.92 | 89.37 |
| Zero-Shot CoT + RCI | **86.22** | 97.24 | 89.88 | 85.83 | 90.94 |
| Few-Shot CoT | 80.31 | 98.82 | 89.37 | 83.46 | 91.73 |
| Few-Shot CoT + RCI | 84.25 | **99.21** | **90.55** | **87.40** | 93.70 |

Table 2: Chain-of-Thought prompting exhibits a synergistic effect when coupled with RCI prompting in arithmetic reasoning tasks.

## 3.2 Computer tasks

### 3.2.1 Setup

**MiniWoB++ benchmark suite.** The miniwob++ task suite is selected as the main benchmark to evaluate our computer agent. MiniWoB++ [36], an extension of MiniWoB [61], is a web-based simulation environment that offers a diverse range of computer tasks, from simple button-clicking to complex compositional tasks requiring advanced reasoning, such as solving math problems. Its shared action space, including keyboard and mouse, and a common state space centered around HTML code enables our proposed agent to be thoroughly evaluated in ample tasks. Additionally, the varying levels of complexity between tasks enable a systematic evaluation of our work. The action space consists of two operations each of which controls the keyboard and mouse. The first action enables typing of arbitrary characters or special keys such as Backspace and Enter. The second action involves moving and clicking the mouse, allowing the agent to interact with visible HTML elements on a webpage. All actions can be executed through natural language instructions defined by regular expressions that are presented within the initial prompts provided to the LLMs. The regular expressions employed in our evaluation are presented in Appendix D. Our action space definition is similar to previous works,

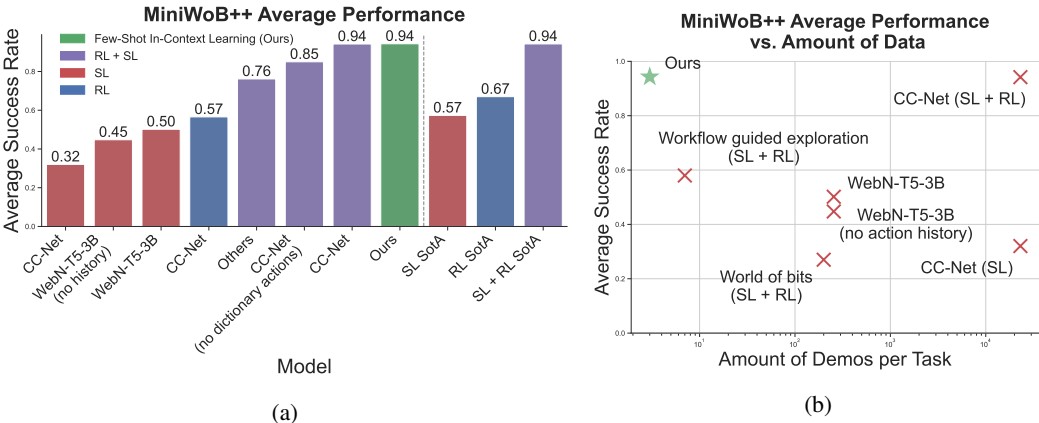

Figure 4: (a) Average performance comparison with baselines. Our agent with RCI prompting achieves state-of-the-art performance in MiniWoB++ environment. The tasks that were included in the averaging process are indicated in Table 18. (b) Relationship between performance and amount of expert training data. Our agent displays comparable performance to the current state-of-the-art scores on the MiniWoB++ benchmark, despite using the least amount of data.

such as [25, 32, 36], in which clicking actions directly interact with HTML elements. However, for typing actions, we extend beyond simple form-filling by using keyboard-based typing actions. Instead of relying on dictionary-based typing actions [30], where the agent simply chooses from a predefined dictionary of texts, our approach requires the agent to predict the proper text input. Our approach, therefore, has a better generalization capability for diverse computer tasks. The state space of our agent consists solely of HTML code.

**Model choices.** For the purpose of evaluating the effectiveness of RCI prompting, multiple language models are used in our experiments. Specifically, we employ three models, namely, GPT-3 (*davinci*) [5], InstructGPT-3 (*text-davinci-002*) [47, 72, 57], and InstructGPT-3 + RLHF (*gpt-3.5-turbo*, *gpt-4*) [47]. Unless otherwise specified, we primarily evaluate our computer agent with the InstructGPT-3 + RLHF models (*gpt-3.5-turbo*, *gpt-4*). Additionally, we use GPT-3 and InstructGPT-3 models for ablation studies. All the models were obtained through the OpenAI API, and further details can be found in Appendix C.1.

**Evaluated tasks.** We employ a set of 55 tasks to enable fair comparisons with baselines, as previous works are only evaluated on a subset of tasks consistently. Furthermore, to assess the performance of models on challenging tasks, we have selected tasks that involve free-form language typing actions, which have been reported to have an almost-zero success rate in previous works (*e.g.,* terminal). Notably, certain commonly evaluated tasks in prior works are excluded due to the excessive length of HTML code for some UI components, which are described in Appendix C.3.

**Metrics** Consistent with prior studies, our main evaluation criterion is the success rate, which measures the ability of our agent to actually complete the assigned task. This rate is calculated as the proportion of successful episodes, which are defined as those in which the agent receives a positive reward. We identified two modes of failure: the production of unexecutable actions and task failure. When the agent generates an unexecutable action following the implicit RCI step, it fails immediately. Moreover, an episode is considered unsuccessful when the agent, despite effectively executing the plan generated, is unable to accomplish the task and thus receives no reward.

### 3.2.2 Outperforming baselines on MiniWoB++ task suite

We present Figure 4a which summarizes the average success rate of our agent and baseline models over the MiniWoB++ benchmark. The results demonstrate significant outperformance of our approach over supervised learning models. Specifically, we observe a 41% higher score than the *WebN-T5-3B*, which employs a finetuned large language model with 12K expert demonstration data. Our approach also outperforms reinforcement learning approaches that require an order of magnitude

more interactions with the environment. Among all the baselines, our approach achieves the second highest score. The sole model that surpasses our agent is the *CC-Net*, which involves co-training of reinforcement learning and imitation learning. However, a direct comparison with *CC-Net* is not possible since it uses dictionary-based typing actions. In other words, *CC-Net* selects text from a predefined list for typing actions in some tasks, while our approach is fully generative. Thus, *CC-Net (without dictionary-based action)* in Figure 4a serves as our appropriate comparison and we outperform it by 6%. The performance data for *CC-Net (with no dictionary-based action)* is obtained from the ablation study section in their paper [30].

Another comparative analysis is performed to evaluate the performance of our agent in contrast to the state-of-the-art agents in three categories, namely supervised learning, reinforcement learning, and a combination of both. To facilitate a fair comparison, we specifically isolate LLM-based state-of-the-art approaches, which share similarities with our approach to solving computer tasks. The best per-task performance achieved by each category is then aggregated, and the outcomes are presented as SotA in Figure 4a. The result shows that our agent surpasses SotA by 37 percentage points in supervised learning and by 27 percentage points in reinforcement learning. Notably, our proposed RCI prompting method outperforms the SotA LLM approach [24], even when the latter employs both finetuning and few-shot examples in prompts. This outcome highlights the effectiveness of our approach in extracting vital knowledge for computer tasks from language models. Our agent even achieves a slight edge over SotA (less than 1 percentage point) in the combined use of supervised and reinforcement learning, which employs significantly more expert data and online interactions. We also provide task-level performance comparisons in Figure 10, where tasks are arranged in ascending order based on the difference between our agent's performance and the baseline. We observed three main failure modes of our agent: (i) underperformance in tasks that require long-horizon planning (*e.g.,* guess-number, search-engine, use-spinner), (ii) difficulty in selecting appropriate actions for tasks that require multi-step reasoning (*e.g.,* tic-tac-toe, use-autocomplete), and (iii) lower scores in tasks that rely on visual rendering of HTML code to solve the task (*e.g.,* count-shape). These failures are explained in more detail in Appendix F.

### 3.2.3 Lowest sample complexity

Figure 4b provides a comparative analysis of the total number of samples used in several models and their mean performance. We begin by discussing *CC-Net* [30] model, which employs 2.4 million expert demonstrations (equivalent to 6,300 hours) collected from 77 human participants across 104 tasks for behavior cloning. This amounts to an average of 23,076 demonstrations per task. In contrast, the *WebN-T5-3B* [24] model uses 12,000 expert demonstrations to fine-tune its pre-trained T5 model. Rather than directly updating model parameters with demonstration data, our approach involves integrating two to three demonstrations into the prompt for in-context learning, which biases the model output without any parameter updates. This approach allows our agent to generalize to unseen tasks with only a handful of demonstrations. Our results show that our agent achieved a higher success rate than all baselines, requiring 120x fewer samples than *WebN-T5-3B* and 11,000x fewer samples than *CC-Net*. Given the challenges of obtaining expert demonstrations for computer tasks, our findings demonstrate the practicality of our approach in automating such tasks.

### 3.2.4 Ablating the groundings

This section examines the impact of grounding improvement on task success rates. We conduct ablations to isolate the contributions of task, state, and agent grounding improvements by eliminating RCI prompting at each stage. We categorize tasks by three different difficulty levels to provide a more detailed understanding of the effects of grounding improvements across a diverse range of tasks. We conducted a task grounding ablation by eliminating the plan sampling stage. This modification entails generating actions directly from the state, without the need for conditioning on a step-by-step plan. State grounding is evaluated by directly applying the agent-grounding update to task-grounded actions. Lastly, we ablate the implicit RCI of the agent grounding by letting the state-grounded action be the final output of the agent. Figure 5 illustrates the performance degradation resulting from each ablation of grounding. Our results indicate that each grounding contribution is essential to solving computer tasks, with each contributing almost equally to the overall success rate. The reason for this is partially due to the fact that the three methods of improving grounding are not mutually exclusive, but rather complementary, with one enhancement in grounding contributing to multiple action groundings. Examples of cross-grounding improvement are provided in Appendix E.

Moreover, it has been observed that state grounding plays a crucial role in enabling an agent to use relevant information during episodes, particularly in scenarios where the initial state does not offer sufficient information to accomplish the task, such as *terminal* task. Interestingly, task grounding significantly improves the success rate when a task requires a long-horizon action plan, such as the *click checkboxes large* task. We also observe that agent grounding significantly enhances the feasibility of actions. Notably, in simpler tasks, the success rate decreases by 60% in contrast to the baseline without the agent grounding. This finding is of particular significance as it distinguishes our work from prior investigations [1, 28], which employ additional trained model components. In contrast, our study solely relies on the reasoning ability of language models.

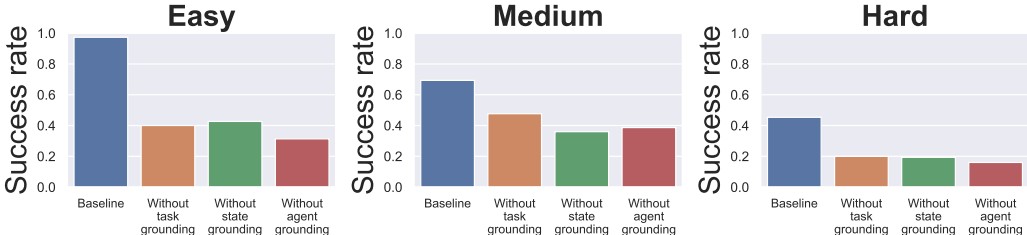

Figure 5: Ablation analysis on the different types of grounding across tasks with varying degrees of difficulty. The experimental design employs the use of InstructGPT-3 + RLHF model (*gpt-3.5-turbo*).

### 3.2.5 Ablating the language model

The performance of our agent is contingent on the quality of the underlying pre-trained language models used, so enhancing language models can lead to an improvement in the agent's performance. In this section, we present a comparison of the agent's performance using three distinct language models: GPT-3, InstructGPT-3, and InstructGPT-3 + RLHF (*gpt-3.5-turbo*). Our objective is to investigate the relationship between LLMs' capability and their ability to solve MiniWoB++ tasks. The experimental setting employed in Section 3.2.4 is replicated in this study. Figure 6 depicts the average success rate of three language models on tasks of varying difficulty levels. Our results reveal that LLMs struggle to effectively complete tasks without instruction fine-tuning. This may be attributed to the absence of intricate prompt engineering, as our observations have indicated that GPT-3 displays sufficient competence in comprehending HTML code, regular expressions, and engaging in reasoning.

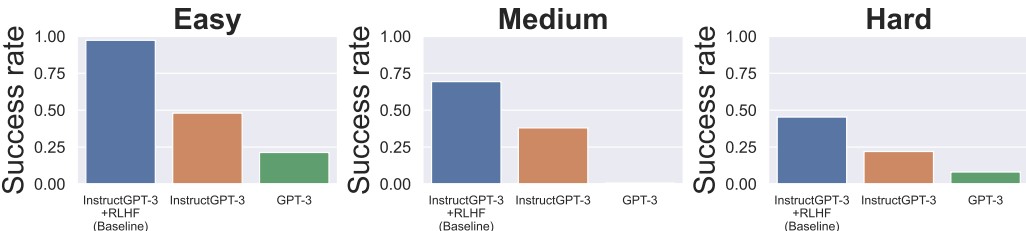

Figure 6: Ablation study on different language models across tasks of varying degrees of difficulty.

## 4 Limitations

In the course of our work, several limitations became apparent that may serve as potential avenues for further research. One central concern is our primary focus on the InstructGPT-3 + RLHF models (*gpt-3.5-turbo*, *gpt-4*), leaving the generalization ability of RCI to other models unexplored. The versatility of RCI across diverse models remains a pertinent question, suggesting that future studies should expand their scope to determine the robustness and adaptability of RCI. Handling lengthy HTML presents another challenge. The current model grapples with extensive HTML states. While it has been suggested that efficiency might be bolstered by pruning HTML states to exclude non-critical

elements, the task itself is non-trivial. A fundamental constraint of LLMs is the limited context length, which can hamper handling extensive HTML states effectively. Addressing this may require architectural adjustments or novel parsing methods. Our agent's action space, mainly restricted to clicks and typing, limits its web navigation capabilities. There's a need to diversify its actions for a more seamless experience. Furthermore, The agent's focus on short-term decisions overlooks the necessity for long-term strategy, especially in tasks requiring coordinated sequences. Broadening this focus is essential for versatile applications. Lastly, the intricate UI components populating contemporary websites present a challenge for LLMs to fully understand the HTML states. The subtle nuances of such components, which may not be discernible through HTML alone, underscore the need for adding more modalities to the state definition. Addressing these issues is crucial to enhance the RCI agent, making it more adaptable and efficient in practical applications.

## 5 Discussion

This work is part of a growing literature showing that LLMs might be all you need for hard decision-making problems [76]. In contrast to imitation learning and reinforcement learning approaches, LLMs can solve novel tasks in a zero-shot or few-shot manner, and don't require task-dependent expert data or a reward function. Furthermore, we expect that as the capabilities of LLMs and foundation models increase, our method will naturally improve as well. However, we find that current capabilities of LLMs aren't as powerful as task-dependent SL+RL approaches on some computer tasks. Also, RCI is more expensive to run compared to approaches that just sample once from the LLM. There are many avenues for future research in increasing the capacity of LLMs in decision-making tasks. First, our experiments use LLMs on HTML code, but ideally methods based on multimodal foundation models [16, 55, 2, 46] will be able to take actions based on text, images, audio, and video as input [4, 18, 44, 71]. Second, the results presented in this paper all use pre-trained LLMs. We expect the performance of our method to increase when using LLMs fine-tuned to solve computer tasks.

Importantly, current LLMs are poor at reasoning tasks, such as playing tic-tac-toe, because they do not think ahead. Although RCI improves reasoning capabilities in LLMs, there exists much work to be done on increasing the reasoning capabilities in LLMs. This will be crucial to accomplish hard cognitive tasks on computers that require thinking ahead. Similar to other prompting-based approaches for reasoning in LLMs, RCI can be viewed as using the LLM's output to write to an external memory, which is later retrieved to choose an action. LLMs with memory have been demonstrated to be computationally universal [60], meaning that in principle all that is needed to run arbitrary programs is the right prompt. Since RCI represents a basic version of this powerful framework, we anticipate the development of more advanced RCI variations in the future. There is a vast array of potential methods that repeatedly feed the output of particular prompts into the LLM. For example, multiple different LLMs can simulate the information exchange between team members in an organization. This would enable the merging of diverse perspectives to tackle complex problems. In such a context, incorporating game theory and multi-agent systems research could significantly enhance the overall performance. Reinforcement learning could be used to discover effective structures involving loops and prompts [81], either through human feedback or a given reward function. This optimization process can be further refined by exploring the space of potential loop and prompt structures, identifying those that yield the best results, and fine-tuning the model accordingly [75].

## Acknowledgement

This material is based upon work supported by the National Science Foundation under Grant #2127309 to the Computing Research Association for the CIFellows 2021 Project.

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

# Appendix

## A Broader Impacts

Although the results presented in this paper are only on a research benchmark, if we extrapolate forward the capabilities of these models and methods, we anticipate vast broader impacts that have the potential to revolutionize numerous industries. By allowing LLMs to execute tasks on computers, our approach can enhance the capabilities of AI assistants and automation tools. This could lead to increased efficiency, reduced labor costs, and improved user experiences across any sector which uses computers to do work. We are most excited about gains in productivity in science and education, including AI research, which will lead to even faster development of new beneficial technologies and treatments.

However, there are many potential misuses and unintended consequences associated with allowing these models to take actions in the world. Malicious actors may leverage LLMs to automate cyber-attacks, manipulate information, or propagate disinformation on a large scale. Additionally, the potential loss of jobs due to widespread automation could lead to economic disruption and increased income inequality. There are also obvious security risks of running LLMs on computers (or even virtual machines) such as prompt injection attacks. Perhaps most alarming, future LLMs taking actions on computers may lead to catastrophic runaway chains of events, especially if LLMs are integrated widely in the economy.

To mitigate these risks, it is crucial for researchers, policymakers, and industry leaders to work together to establish regulations and ethical guidelines that govern the development and deployment of such technologies. Ensuring transparency, accountability, and fairness in AI systems will be vital in harnessing the benefits while minimizing potential harm. We also believe that the time has come where we as a research community must discuss possible ways to coordinate to slow down the pace of developing highly-disruptive technology, if necessary.

# B  Related Works

## B.1  Automated computer tasks

The automation of computer tasks is an important topic for both information retrieval and natural language processing [43, 50, 49, 31, 65]. Recent efforts have focused on the development of reinforcement learning agents that interact with websites using raw mouse and keyboard actions [61]. MiniWoB, a benchmark proposed in [61], has been extended in MiniWoB++ [36], which has become a widely-used platform for studying models for computer tasks. Reinforcement learning and imitation learning have been employed in several studies to tackle MiniWoB++ tasks [36, 25, 32, 23]. However, achieving human-level performance requires a significant amount of expert demonstration data (6,300 hours), as demonstrated in [30]. Recent work [24, 19] has suggested the use of large language models (LLMs) to comprehend HTML code and vision transformer [15] to extract screenshot image features, with a few-shot in-context approach showing promising results without extensive RL exploration. Nevertheless, significant amounts of expert demonstration data are still required to finetune LLMs. On the contrary, the agent we suggest needs less than two demonstrations per task on average and doesn't necessitate any finetuning. WebGPT [42] and WebShop [77] show that LLMs can automate some web-based tasks by introducing a handful of custom commands such as `Search <query>` and `Next Page`. As a result, these methods are limited in scope and do not work on general computer tasks which require keyboard strokes and mouse clicks. In contrast, our approach can tackle open-domain tasks at scale.

## B.2  LLMs with actions

In recent years, there have been significant advancements in large language models (LLMs), leading to new possibilities for utilizing natural language for decision-making tasks. One approach involves augmenting LLMs with executable actions [41]. Huang et al. [28] showed that LLMs can be used to plan and achieve simple household tasks, utilizing a method for grounding the actions generated by LLMs by comparing their embeddings with a predefined list of admissible actions. However, their work did not consider state grounding. Another study by Ahn et al. [1] proposed SayCan, which grounded the actions by multiplying each candidate action's probability under FLAN [72] with the action's value function, serving as an indicator for the suitability of actions. Huang et al. [29] proposed an extension to the SayCan model called Inner Monologue, which incorporates a feedback loop for state grounding. However, Inner Monologue still requires a pre-trained language-conditioned robot policy with underlying reasoning capabilities that are not free-formed and flexible, thereby hindering generalization to diverse task domains. Similarly, Zeng et al.[80] employed a combination of LLMs with a visual-language model (VLM) and a pre-trained language-conditioned robot policy [63] to perform open vocabulary pick-and-place robotic tasks. Meanwhile, Dasgupta et al.[13] used Chinchilla[26] as a planner for an agent in the PycoLab environment, but their actor module requires pre-training with reinforcement learning (RL) to follow natural language instructions. In a related line of research, Carta et al. [7] employed online RL fine-tuning to achieve functional grounding of LLMs in the BabyAI-Text environment. In contrast to these previous approaches, our method does not rely on additional model components beyond LLMs for grounding actions. Instead, we propose the use of RCI prompting, which enables LLMs to update their actions to be grounded autonomously. As a result, our approach can scale to a wider range of action spaces, including keyboard and mouse actions. Furthermore, prior approaches have been limited by the need for fine-tuning. In contrast, our RCI prompting method is a zero-shot approach that overcomes these limitations. More recently, an approach to improve the efficacy of LLMs involves their integration with APIs, allowing them to use external tools such as information retrieval systems, code interpreters, and web browsers [59, 68, 40, 22]. Notably, these external tools necessitate manual engineering and may be constrained in their functionality. In contrast, our agent is equipped with a general computer interface, enabling it to access a wide range of functionalities offered by computers.

## B.3  LLMs with reasoning

Recent research has also demonstrated that large language models (LLMs) exhibit enhanced performance in compositional tasks when they produce traces of the underlying reasoning process along with the final answer, as evidenced by studies such as [73, 45, 33, 52]. This discovery has led to the emergence of a new line of research where reasoning capabilities are used to address tasks

beyond reasoning [78, 29], or enhance reasoning proficiency [33, 37, 70, 39, 79, 53, 75, 66, 14]. Furthermore, various reasoning architectures have been proposed, expanding from naive prompting, such as Selection-Inference [11], Least-to-Most [82], and Faithful reasoning [10]. In the existing literature, a work closely related to our research is ReAct [78] which interleaves reasoning and action for resolving the issue of hallucination and error propagation as well as helping the model induce, track, and update action plans. An alternative method, Reflexion [62], extends ReAct by improving its performance by allowing LLMs to consider previous trial and error experiences. Nevertheless, due to the necessity of multiple rounds of explicit task-specific success feedback from trial and error, this approach may not scale as effortlessly as ours because it requires task-specific success feedback. Similarly, Corrective Re-prompting, as proposed by Raman et al. [54] necessitates the establishment of task-specific preconditions. RCI pertains to an extended reasoning architecture where LLMs are instructed to find errors in their outputs and improve them accordingly, which can further be used to ground actions generated from LLMs in decision-making problems. Saunders et al. [58] used a similar approach to ours by utilizing the self-critiquing ability of LLMs to generate critical feedback on summaries produced by LLMs. The aim is to accelerate the human evaluation process by uncovering possible errors in the generated summaries. Likewise, Ganguli et al. [20], employed LLMs to morally self-correct their outputs to prevent the generation of harmful content. The most recent work that is in the same vein with RCI is Self-Refine [38] which uses localized and aspect-based feedback to iteratively refine outputs from LLMs. However, our work is, to the best of our knowledge, the first to demonstrate the self-critiquing capability of LLMs only with implicit feedback (*e.g.,* "Find problems with this plan") in enhancing reasoning proficiency.

# C  Experimental setup

## C.1  Language models

In our evaluation, various pre-trained language models were used. RCI prompting on reasoning tasks is evaluated using *gpt-3.5-turbo*, which is presented in Table 1 and Table 2. Our primary evaluation on MiniWoB++ tasks is conducted using *gpt-3.5-turbo* and *gpt-4*, as shown in Figure 4a and Figure 10. We also used *davinci*, *text-davinci-002*, and *gpt-3.5-turbo* for our ablation study on MiniWoB++ tasks. For all model usage, a maximum token length of 256 and a temperature value of 0, indicating greedy decoding, are used. All models are accessed through the OpenAI API between January 2023 and March 2023.

| Language model | # of parameters | Max. tokens | API provider | API name |
|---|---|---|---|---|
| GPT-3 | 175 B($*$1) | 2,049 | OpenAI API | davinci |
| InstructGPT-3 | $-($*$2)$ | 4,097 | OpenAI API | text-davinci-002 |
| InstructGPT-3 + RLHF | $-($*$2)$ | 4,096 | OpenAI API | gpt-3.5-turbo |
| InstructGPT-3 + RLHF | $-($*$2)$ | 8,192 | OpenAI API | gpt-4 |

Table 3: Description of language models. ($*$1) We identify the model size of GPT-3 by referring to the official document that OpenAI provides (https://beta.openai.com/docs/model-index-for-researchers). ($*$2) The size of InstructGPT-based models remains undisclosed by its provider.

## C.2  Reasoning tasks

We conducted an evaluation of RCI prompting on eight datasets, encompassing two categories of reasoning tasks: arithmetic and commonsense. In the domain of arithmetic reasoning, we considered six datasets: SingleEq [34], AddSub [27], MultiArith [56], AQuA [35], GSM8K [9], and SVAMP [51]. For commonsense reasoning, we utilized the CommonsenseQA dataset [67] and the StrategyQA dataset [21]. To ensure a fair comparison with baselines, we specifically selected tasks that were employed in the work of Kojima et al. [33]. In the experiment on reasoning tasks, we enable RCI loop to get implicit feedback to correct outputs. We fix the maximum number of loops to 2. Following previous works [38, 62, 74], we use the correct label to decide when to stop the RCI loop. In our setting, we can consider the correct label to be another source of feedback (label feedback).

## C.3  MiniWoB++ task selection

In order to ensure a fair and comprehensive evaluation, a subset of MiniWoB++ tasks we use in the evaluation is selected from the evaluation of *WebN-T5-3B* [24], the most recent work on MiniWoB++ tasks, which employs LLMs. However, certain tasks such as book-flight, choose-date-easy, choose-date-medium, choose-date, and click-pie have been excluded from our evaluation due to their HTML code exceeding the maximum context length of language models. On the other hand, some of the challenging tasks such as terminal and simple-algebra have been included in the evaluation. The choice of these tasks is determined by the suboptimal performance of *CC-Net* [30], which currently represents the state-of-the-art model in the field. The purpose of this inclusion is to showcase the potential of leveraging LLMs in computer tasks, in contrast to the conventional approaches of Supervised Learning (SL) and Reinforcement Learning (RL). While our agent has not been evaluated on tasks that necessitate additional actions, such as drag and copy & paste, we posit that their inclusion can be readily achieved through the expansion of the actions space specification within the prompts.

## C.4  MiniWoB++ task selection for ablation studies

In ablation studies, we categorize the tasks based on the success rate achieved by our agent with *gpt-3.5-turbo*. We select a subset of tasks from three levels of difficulty, as depicted in Table 4.

| | | | |
|---|---|---|---|
| easy [0.9, 1] | | click-shape | 0.98 |
| | | click-widget | 0.98 |
| | | enter-date | 0.96 |
| medium [0.6, 0.9) | | click-checkboxes-soft | 0.72 |
| | | click-collapsible-2 | 0.62 |
| | | click-tab-2 | 0.74 |
| hard [0, 0.6) | | click-tab-2-hard | 0.56 |
| | | count-shape | 0.4 |
| | | guess-number | 0.2 |

Table 4: The tasks used in the ablation study are classified according to their level of difficulty.

## C.5 Modifications on MiniWoB++ tasks

In Table 5, we outline several modifications that were incorporated into the MiniWoB++ benchmark for the purpose of our evaluation with language models that have a limited context length.

| Tasks | Modifications |
|---|---|
| social-media-all
social-media
social-media-some | We constrain the quantity of media components ranging from three to six. |
| email-inbox-forward-nl-turk
email-inbox-forward-nl
email-inbox-nl-turk | The quantity of randomly generated emails has been restricted to a range of three to six. |

Table 5: Modifications on MiniWoB++ tasks.

# D    Prompts for MiniWoB++ tasks

We have an autonomous computer control agent that can perform a set of instructions to control computers.
First, given the instruction that matches the regular expression, <type regex>, it can type a list of characters via the keyboard. This instruction should specify the target keyboard input for the agent to type. Before this typing instruction, you should first locate the cursor by clicking the input box with the click instruction.
Second, given the instruction that matches the regular expression, <press regex>, it can press a specific key on the keyboard.
Third, given the instruction that matches the regular expression, <clickoption regex>, it can click an option HTML element in a list with an XPath that is visible on the webpage. The target of this instruction should be a valid XPath.
Fourth, given the instruction that matches the regular expression, <movemouse regex>, it can move the mouse cursor on an HTML element with an XPath that is visible on the webpage.
Lastly, given the instruction that matches the regular expression, <clickxpath regex>, it can click an HTML element with an XPath that is visible on the webpage. The target of this instruction should be a valid XPath.

Table 6: Agent specification.

```
<type regex> = "^type\s.{1,}$"
<press regex> = "^press\s(enter|arrowleft|arrowright|arrowup|arrowdown|
    backspace)$"
<clickoption regex> = "^clickoption\s.{1,}$"
<movemouse regex> = "^movemouse\s.{1,}$"
<clickxpath regex> = "^clickxpath\s.{1,}$"
```

Listing 1: Regular expressions for specifying the admissible actions.

Based on the current plan and task, the next proper instruction should be `

Table 7: Task grounding.

Considering the output on the webpage, the specific instruction should be `

Table 8: State grounding.

This action does not match the regular expressions. The updated instruction that matches one of the regular expressions is `

Table 9: Agent grounding.

# E    Examples of cross grounding

---

task: click on the "ok" button on the webpage.
plan:
1. type "sit arcu nullam:"
2. clickxpath //button[@data-wob_ref="6"]

We have a history of instructions that have been already executed by the autonomous agent so far.
No instruction has been executed yet.

Based on the current plan and the task, the first instruction should be `type "sit arcu nullam:"`.
Therefore, considering the output on the webpage, this instruction for solving the task should be `clickxpath //button[@data-wob_ref="6"]`.

---

Table 10: The state grounding optimizes the action plan. In the initial plan, the inclusion of the type instruction is deemed superfluous for accomplishing the task.

---

task: Find the email by Clarinda and reply to them with the text "Vel id at.".
plan:
1. clickxpath //div[@class='email-sender' and text()='Clarinda']
2. clickxpath //span[@class='email-reply']
3. clickxpath //*[@id="reply-text"]
4. type Vel id at.
5. clickxpath //*[@id="send-reply"]

We have a history of instructions that have been already executed by the autonomous agent so far.
0: clickxpath //div[@class='email-sender' and text()='Clarinda']
1: clickxpath //span[@class='email-reply']
2: clickxpath //*[@id="reply-text"]

Based on the current plan and the task, the next proper instruction after the history should be `type Vel id at`. Therefore, considering the output on the webpage, this instruction should be `type Vel id at.`

---

Table 11: The state grounding corrects actions to be grounded in the task. Although the target of typing action in the action plan is "Vel id at." the generated task-grounded action lacked the last punctuation mark. Nonetheless, the state grounding update was successful in remedying this discrepancy, resulting in the state-grounded action being "Vel id at." Consequently, the state-grounded action now matches the typing action specified in the action plan.

# F  Failure modes on MiniWoB++ tasks

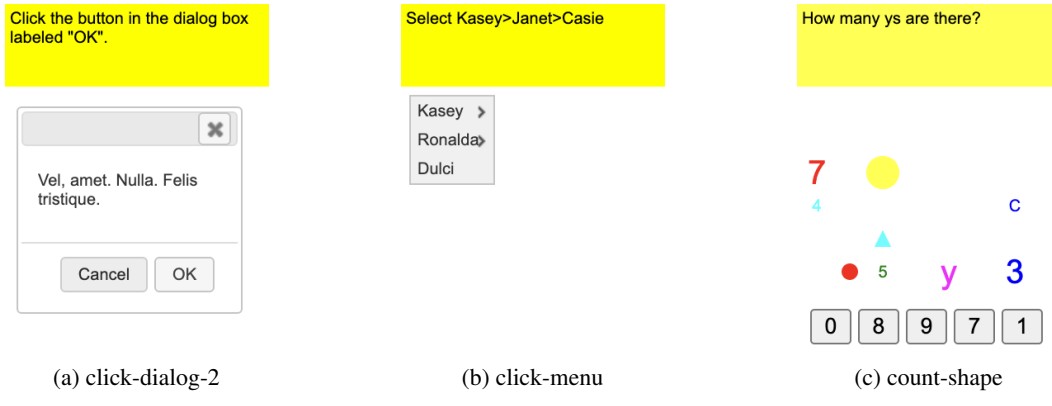

(a) click-dialog-2            (b) click-menu            (c) count-shape

Figure 7: Screenshots.

## F.1  click-dialog-2

The state-grounding process is occasionally inadequate for maintaining task-grounding of actions. Specifically, in the click-dialog-2 task, grounding actions in the current state results in task-irrelevant actions. Consequently, we opt to exclude the state grounding step for this particular task.

---

task: click the button in the dialog box labeled "x".
plan:
1. Clickxpath //div[@class='ui-dialog-titlebar ui-corner-all ui-widget-header ui-helper-clearfix ui-draggable-handle']/button[@title='Close']

We have a history of instructions that have been already executed by the autonomous agent so far.
No instruction has been executed yet.

Based on the current plan and task, the first instruction should be `clickxpath //div[@class='ui-dialog-titlebar ui-corner-all ui-widget-header ui-helper-clearfix ui-draggable-handle']/button[@title='Close']`. Therefore, considering the output on the webpage, this instruction should be ` clickxpath //div[@class='ui-dialog-buttonset']/button[@title='OK']` to click the button in the dialog box labeled "x".

---

Table 12: Example trajectory on click-dialog-2. The state grounding updates the task-grounded actions to be irrelevant to the task.

## F.2  click-menu

We find that including explanatory trajectories in the few-shot examples can significantly reduce the number of required few-shot examples. Specifically, in the click-menu task, the selection of the target component necessitates the mouse hovering over the expandable menu. Therefore, we provided explanations regarding the mouse hovering technique in the few-shot examples.

## F.3  count-shape

This task necessitates an agent to visually perceive rendered images on a canvas HTML element and enumerate the relevant shapes. Unfortunately, our agent encounters difficulties in tackling such tasks which entail the need for perceiving visual representations, as it solely depends on HTML observation.

| |
|---|
| task: select Leonelle > Jane > Amabel from the HTML code provided.
plan:
1. movemouse //*[text()="Leonelle"] // we need to hover the mouse to expand the menu
2. movemouse //*[text()="Jane"] // we need to hover the mouse to expand the menu
3. clickxpath //*[text()="Amabel"] |
| task: select the option "Joye > Phylis" from the HTML code provided.
plan:
1. movemouse //*[text()="Joye"] // we need to hover the mouse to expand the menu
2. clickxpath //*[text()="Phylis"] |

Table 13: Example trajectory on click-menu. Providing explanatory trajectory enhances the sample-efficiency of few-shot examples.

| |
|---|
| task: answer the question "How many large items are there?".
plan:
1. Clickxpath //button[text()="2"] |

Table 14: Example trajectory on count-shape. The agent struggle to solve tasks that requires visual rendering of HTML.

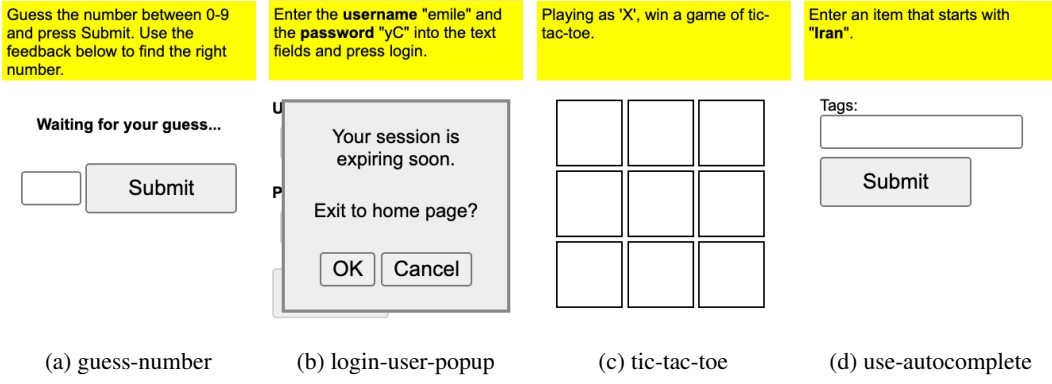

(a) guess-number  (b) login-user-popup  (c) tic-tac-toe  (d) use-autocomplete

Figure 8: Screenshots.

### F.4 guess-number

When considering decision-making that involves a long-term perspective, our agent encounters difficulties in generating actions that are grounded in the current task. For instance, in this guess-number task, the plan generated by our agent involves the sequential typing of numbers from 0 to 9 in order to solve the task. However, due to the high number of steps required (over 50), our agent occasionally produces actions that deviate from the original plan, ultimately resulting in irreparable failure.

### F.5 login-user-popup

This task involves the identification of an optimal strategy for an agent to handle an unpredictable pop-up window that emerges during a task. Due to the unexpected nature of the pop-up window, pre-planning the closure of the window is not feasible. Our agent is designed to adapt the agent's actions to the current state, so it should generate an appropriate instruction to close the pop-up window in the state-grounding step. Nevertheless, there are instances where it is unsuccessful in executing the pop-up window closure, leading to task failure.

task: Guess the number between 0-9 and press Submit. Use the feedback below to find the right number.
plan:
1. clickxpath //*[@id="tt"]
2. type 0
3. clickxpath //*[@id="subbtn"]
4. clickxpath //*[@id="tt"]
5. press backspace
6. type 1
7. clickxpath //*[@id="subbtn"]
8. clickxpath //*
9. press backspace
10. type 2
11. clickxpath //*[@id="subbtn"]
12. clickxpath //*[@id="tt"]
13. press backspace
14. type 3
15. clickxpath //*[@id="subbtn"]
16. clickxpath //*[@id="tt"]
17. press backspace
18. type 4
19. clickxpath //*[@id="subbtn"]
20. clickxpath //*[@id="tt"]
21. press backspace
22. type 5
23. clickxpath //*[@id="subbtn"]
24. clickxpath //*[@id="tt"]
25. press backspace
26. ...

We have a history of instructions that have been already executed by the autonomous agent so far.
0: clickxpath //*[@id="tt"]
1: type 0
2: clickxpath //*[@id="subbtn"]
3: press backspace
4: type 1
5: clickxpath //*[@id="subbtn"]

Based on the current plan and the task, the next proper instruction should be `type 2`.

Table 15: Example trajectory on guess-number. The agent fails to generate task-grounded action when long-horizon decision making is required.

### F.6 tic-tac-toe

We also examine the causes of the poor success rate in playing tic-tac-toe. Two distinct factors are identified as responsible for its failure. The first factor is that our agent is unable to adopt a defensive strategy when its opponent is only one move away from victory and there is no immediate opportunity to win the game. The second factor relates to the inability to consider the possibility of its attack being blocked by the opponent. A case in point is the bottom-left illustration in Figure 9, where our agent's fourth move is ineffective since its diagonal direction is already obstructed by the opponent.

### F.7 use-autocomplete

In use-autocomplete task, our agent demonstrated an ability to select words beginning with specific characters. However, it struggles when it comes to selecting words that ended with particular

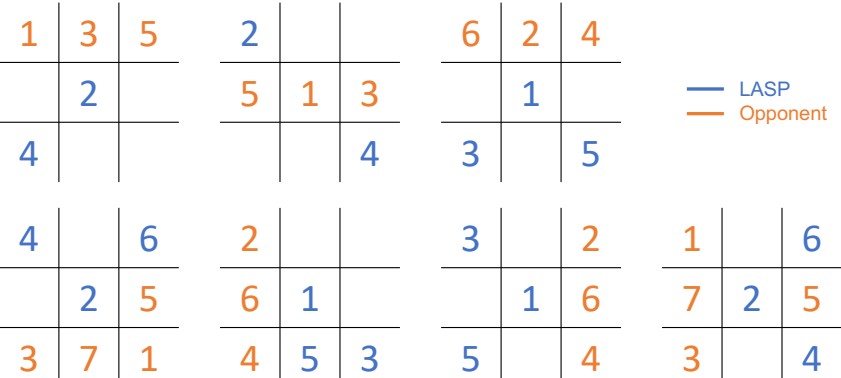

Figure 9: Failure examples of tic-tac-toe task.

characters. This difficulty arose from the need to identify the correct word from the autocomplete list, a process that required the agent to press the down arrow button the exact number of times needed to locate the desired word within the list.

---

task: Enter an item that starts with "Se" and ends with "ia".
plan:
1. click the input field (e.g., clickxpath //*[@id="tags"])
2. type the starting word (e.g., type Se)
3. press the down arrow key to select the word ends with "ia" (e.g., press arrowdown)
5. select the word (e.g., press enter)
6. click the submit button (e.g., clickxpath //*[@id="subbtn"])

---

Table 16: Example trajectory on use-autocomplete. The agent fails to identify how many times it has to press the down-arrow key.

# G Additional results

| | Arithmetic | | | | | | Common Sense | |
|---|---|---|---|---|---|---|---|---|
| | GSM8K | MultiArith | AddSub | SVAMP | SingleEq | AQuA | CommonSenseQA | StrategyQA |
| Zero-Shot | 77.95 | 94.48 | 88.58 | 80.70 | 86.61 | **60.23** | **64.56** | 48.42 |
| Zero-Shot + RCI | 72.83 | 86.61 | 83.07 | 79.13 | 83.07 | 59.84 | 46.06 | **48.81** |
| Zero-Shot CoT | **82.28** | 96.85 | 83.86 | 79.92 | 89.37 | n/a | n/a | n/a |
| Zero-Shot CoT + RCI | 74.40 | 87.79 | 83.07 | 79.13 | 83.46 | n/a | n/a | n/a |
| Few-Shot CoT | 80.31 | **98.82** | **89.37** | **83.46** | **91.73** | n/a | n/a | n/a |
| Few-Shot CoT + RCI | 71.65 | 93.70 | 83.46 | 78.74 | 83.46 | n/a | n/a | n/a |

Table 17: In the absence of external feedback, RCI prompting on reasoning benchmarks exhibits performance equivalent to, or below that of a zero-shot approach.

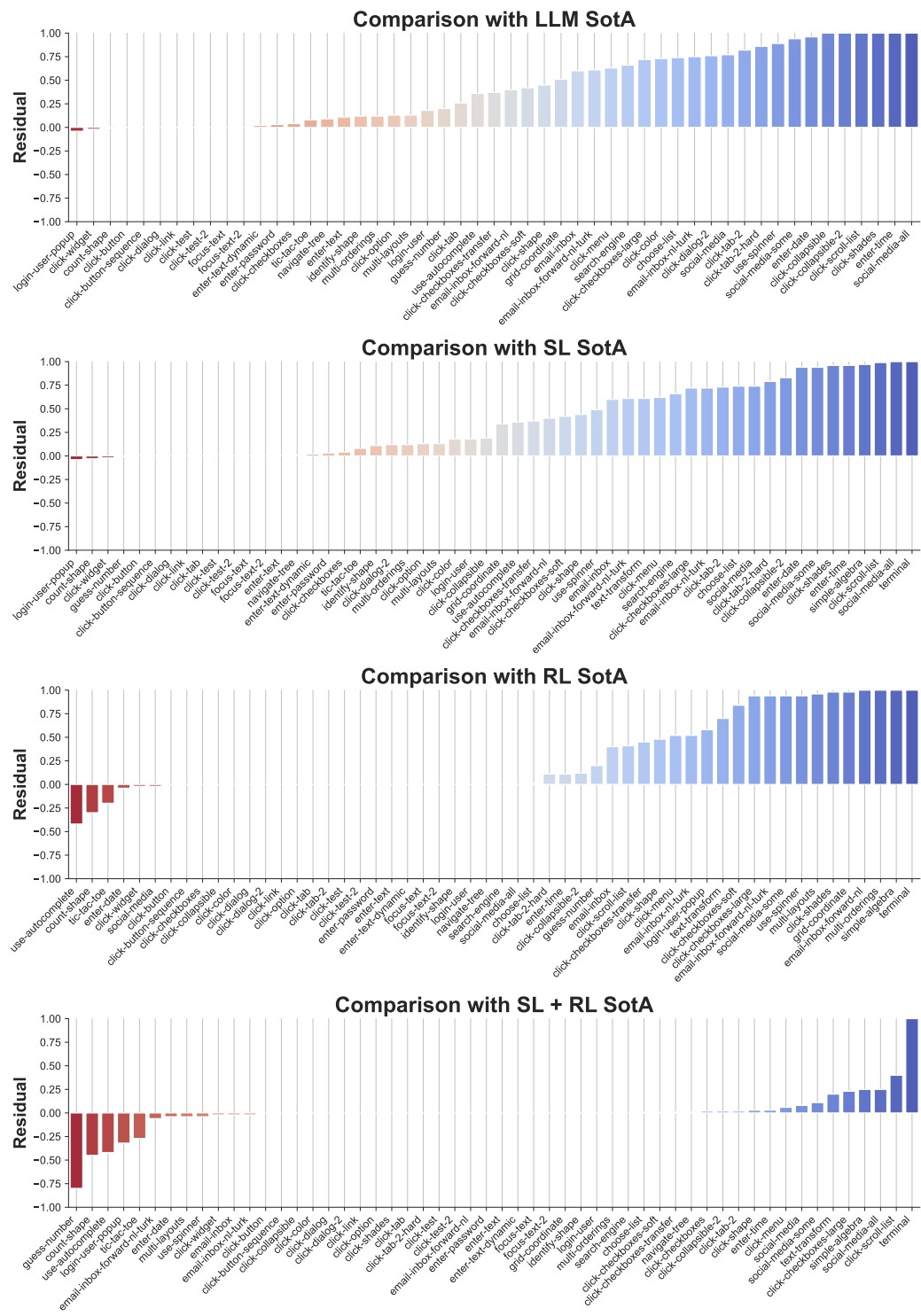

Figure 10: The task-level performance comparison with the state-of-the-art (SotA) baselines. The y-axis represents the residual values, which are obtained by subtracting the performance of SotA from our agent's performance.

| TASK | Ours | Ours (w/ GPT-4) | WebN-T5-3B | CC-Net (SL + RL) | CC-Net (RL) | CC-Net (SL) | Others (SL + RL) | SotA (SL) | SotA (RL) | SotA (SL + RL) |
|---|---|---|---|---|---|---|---|---|---|---|
| bisect-angle | n/a | n/a | n/a | 0.97 | 1.00 | 0.29 | 0.80 | 0.29 | 1.00 | 0.97 |
| book-flight | n/a | n/a | 0.00 | 0.87 | 0.00 | 0.00 | 1.00 | 0.00 | 1.00 | 0.87 |
| chase-circle | n/a | n/a | n/a | 0.93 | 0.93 | 0.80 | 1.00 | 0.80 | 0.93 | 1.00 |
| choose-date-easy | n/a | n/a | 0.03 | 0.99 | 0.99 | 0.42 | n/a | 0.42 | 0.99 | 0.99 |
| choose-date-medium | n/a | n/a | 0.00 | 0.99 | 0.02 | 0.26 | n/a | 0.26 | 0.02 | 0.99 |
| choose-date | n/a | n/a | 0.00 | 0.97 | 0.01 | 0.12 | 1.00 | 0.12 | 1.00 | 0.97 |
| choose-list | 1.00 | 1.00 | 0.26 | 0.99 | 0.99 | 0.19 | 0.26 | 0.26 | 0.99 | 0.99 |
| circle-center | n/a | n/a | n/a | 0.97 | 1.00 | 0.36 | 0.98 | 0.36 | 1.00 | 0.98 |
| click-button-sequence | 1.00 | 1.00 | 1.00 | 1.00 | 1.00 | 0.47 | 1.00 | 1.00 | 1.00 | 1.00 |
| click-button | 1.00 | 1.00 | 1.00 | 1.00 | 0.80 | 0.78 | 1.00 | 1.00 | 1.00 | 1.00 |
| click-checkboxes-large | 0.94 | 0.94 | 0.22 | 0.71 | 0.00 | 0.00 | 0.84 | 0.22 | 0.00 | 0.71 |
| click-checkboxes-soft | 0.72 | 0.96 | 0.54 | 0.95 | 0.12 | 0.04 | 0.94 | 0.54 | 0.12 | 0.95 |
| click-checkboxes-transfer | 1.00 | 1.00 | 0.63 | 0.99 | 0.55 | 0.36 | 0.64 | 0.63 | 0.55 | 0.99 |
| click-checkboxes | 1.00 | 1.00 | 0.96 | 0.98 | 0.45 | 0.32 | 1.00 | 0.96 | 1.00 | 0.98 |
| click-collapsible-2 | 0.62 | 1.00 | 0.00 | 0.98 | 0.88 | 0.17 | 0.99 | 0.17 | 0.88 | 0.98 |
| click-collapsible | 1.00 | 1.00 | 0.00 | 1.00 | 1.00 | 0.81 | 1.00 | 0.81 | 1.00 | 1.00 |
| click-color | 1.00 | 1.00 | 0.27 | 1.00 | 1.00 | 0.82 | 1.00 | 0.82 | 1.00 | 1.00 |
| click-dialog-2 | 1.00 | 1.00 | 0.24 | 1.00 | 1.00 | 0.88 | 1.00 | 0.88 | 1.00 | 1.00 |
| click-dialog | 1.00 | 1.00 | 1.00 | 1.00 | 1.00 | 0.95 | 1.00 | 1.00 | 1.00 | 1.00 |
| click-link | 1.00 | 1.00 | 1.00 | 0.99 | 0.94 | 0.59 | 1.00 | 1.00 | 1.00 | 1.00 |
| click-menu-2 | n/a | n/a | n/a | 0.83 | 0.96 | 0.52 | 0.16 | 0.52 | 0.96 | 0.83 |
| click-menu | 1.00 | 1.00 | 0.37 | 0.94 | 0.48 | 0.22 | 0.13 | 0.38 | 0.48 | 0.94 |
| click-option | 1.00 | 1.00 | 0.87 | 0.99 | 0.78 | 0.21 | 1.00 | 0.87 | 1.00 | 1.00 |
| click-pie | n/a | n/a | 0.51 | 0.97 | 0.92 | 0.15 | 1.00 | 0.51 | 1.00 | 0.97 |
| click-scroll-list | 1.00 | 1.00 | 0.00 | 0.60 | 0.59 | 0.01 | 0.07 | 0.01 | 0.59 | 0.60 |
| click-shades | 1.00 | 1.00 | 0.00 | 1.00 | 0.02 | 0.04 | 0.99 | 0.04 | 0.02 | 1.00 |
| click-shape | 0.98 | 0.98 | 0.53 | 0.95 | 0.50 | 0.11 | 0.64 | 0.54 | 0.50 | 0.95 |
| click-tab-2-easy | n/a | n/a | n/a | 0.99 | 0.94 | 0.61 | n/a | 0.61 | 0.94 | 0.99 |
| click-tab-2-hard | 0.76 | 0.98 | 0.12 | 0.98 | 0.87 | 0.19 | n/a | 0.19 | 0.87 | 0.98 |
| click-tab-2-medium | n/a | n/a | n/a | 0.99 | 0.96 | 0.54 | n/a | 0.54 | 0.96 | 0.99 |
| click-tab-2 | 0.74 | 1.00 | 0.18 | 0.98 | 0.91 | 0.27 | 1.00 | 0.27 | 1.00 | 0.98 |
| click-tab | 1.00 | 1.00 | 0.74 | 1.00 | 1.00 | 0.95 | 1.00 | 1.00 | 1.00 | 1.00 |
| click-test-2 | 1.00 | 1.00 | 1.00 | 1.00 | 1.00 | 0.95 | 1.00 | 1.00 | 1.00 | 1.00 |
| click-test-transfer | n/a | n/a | n/a | 1.00 | 1.00 | 0.94 | n/a | 0.94 | 1.00 | 1.00 |
| click-test | 1.00 | 1.00 | 1.00 | 1.00 | 1.00 | 1.00 | 1.00 | 1.00 | 1.00 | 1.00 |
| click-widget | 0.98 | 0.98 | 1.00 | 1.00 | 1.00 | 0.56 | 1.00 | 1.00 | 1.00 | 1.00 |
| copy-paste-2 | n/a | n/a | n/a | 0.63 | 0.00 | 0.01 | 0.00 | 0.01 | 0.00 | 0.63 |
| copy-paste | n/a | n/a | n/a | 0.79 | 0.00 | 0.04 | 0.00 | 0.04 | 0.00 | 0.79 |
| count-shape | 0.40 | 0.40 | 0.41 | 0.85 | 0.70 | 0.21 | 0.76 | 0.43 | 0.70 | 0.85 |
| count-sides | n/a | n/a | n/a | 1.00 | 1.00 | 0.74 | 0.30 | 0.74 | 1.00 | 1.00 |
| drag-box | n/a | n/a | n/a | 1.00 | 0.19 | 0.61 | 0.31 | 0.61 | 0.19 | 1.00 |
| drag-cube | n/a | n/a | n/a | 0.79 | 0.95 | 0.23 | 0.18 | 0.23 | 0.95 | 0.79 |
| drag-item | n/a | n/a | n/a | 1.00 | 0.00 | 0.61 | n/a | 0.61 | 0.00 | 1.00 |
| drag-items-grid | n/a | n/a | n/a | 0.98 | 0.00 | 0.05 | 0.01 | 0.05 | 0.00 | 0.98 |
| drag-items | n/a | n/a | n/a | 0.99 | 0.00 | 0.13 | 0.41 | 0.13 | 0.00 | 0.99 |
| drag-shapes | n/a | n/a | n/a | 0.99 | 0.23 | 0.26 | 0.92 | 0.26 | 0.23 | 0.99 |
| drag-sort-numbers | n/a | n/a | n/a | 0.97 | 0.00 | 0.11 | 0.66 | 0.11 | 0.00 | 0.97 |
| email-inbox-delete | n/a | n/a | n/a | 1.00 | 1.00 | 0.22 | 1.00 | 0.22 | 1.00 | 1.00 |
| email-inbox-forward-nl-turk | 0.94 | 0.94 | 0.33 | 1.00 | 0.00 | 0.00 | n/a | 0.33 | 0.00 | 1.00 |
| email-inbox-forward-nl | 1.00 | 1.00 | 0.60 | 1.00 | 0.00 | 0.00 | n/a | 0.60 | 0.00 | 1.00 |
| email-inbox-forward | n/a | n/a | n/a | 1.00 | 0.00 | 0.01 | n/a | 0.01 | 0.00 | 1.00 |
| email-inbox-important | n/a | n/a | n/a | 1.00 | 1.00 | 0.30 | n/a | 0.30 | 1.00 | 1.00 |
| email-inbox-nl-turk | 0.98 | 0.98 | 0.23 | 1.00 | 0.46 | 0.05 | 0.93 | 0.26 | 0.46 | 1.00 |
| email-inbox-noscroll | n/a | n/a | n/a | 1.00 | 0.48 | 0.13 | n/a | 0.13 | 0.48 | 1.00 |
| email-inbox-reply | n/a | n/a | n/a | 1.00 | 0.00 | 0.00 | n/a | 0.00 | 0.00 | 1.00 |
| email-inbox-star-reply | n/a | n/a | n/a | 1.00 | 0.47 | 0.11 | n/a | 0.11 | 0.47 | 1.00 |
| email-inbox | 0.98 | 0.98 | 0.38 | 1.00 | 0.58 | 0.09 | 0.99 | 0.38 | 0.58 | 1.00 |
| enter-date | 0.96 | 0.96 | 0.00 | 1.00 | 1.00 | 0.02 | 1.00 | 0.02 | 1.00 | 1.00 |
| enter-password | 1.00 | 1.00 | 0.97 | 1.00 | 0.01 | 0.02 | 1.00 | 0.97 | 1.00 | 1.00 |
| enter-text-2 | n/a | n/a | n/a | 0.98 | 0.00 | 0.04 | 0.00 | 0.04 | 0.00 | 0.98 |
| enter-text-dynamic | 1.00 | 1.00 | 0.98 | 1.00 | 1.00 | 0.39 | 1.00 | 0.98 | 1.00 | 1.00 |
| enter-text | 1.00 | 1.00 | 0.89 | 1.00 | 1.00 | 0.35 | 1.00 | 0.99 | 1.00 | 1.00 |
| enter-time | 1.00 | 1.00 | 0.00 | 0.97 | 0.89 | 0.04 | 0.90 | 0.04 | 0.89 | 0.97 |
| find-midpoint | n/a | n/a | n/a | 0.97 | 0.97 | 0.35 | 0.31 | 0.35 | 0.97 | 0.97 |
| find-word | n/a | n/a | n/a | 0.88 | 0.00 | 0.05 | 0.00 | 0.05 | 0.00 | 0.88 |
| focus-text-2 | 1.00 | 1.00 | 1.00 | 1.00 | 1.00 | 0.96 | 1.00 | 1.00 | 1.00 | 1.00 |
| focus-text | 1.00 | 1.00 | 1.00 | 1.00 | 1.00 | 0.99 | 1.00 | 1.00 | 1.00 | 1.00 |
| grid-coordinate | 1.00 | 1.00 | 0.49 | 1.00 | 0.02 | 0.66 | 1.00 | 0.66 | 0.02 | 1.00 |
| guess-number | 0.20 | 0.20 | 0.00 | 1.00 |  | 0.21 | 0.20 | 0.21 | 0.00 | 1.00 |
| highlight-text-2 | n/a | n/a | n/a | 1.00 | 0.34 | 0.40 | 0.13 | 0.40 | 0.34 | 1.00 |
| highlight-text | n/a | n/a | n/a | 1.00 | 1.00 | 0.51 | 0.90 | 0.51 | 1.00 | 1.00 |
| identify-shape | 0.76 | 1.00 | 0.88 | 1.00 | 1.00 | 0.68 | 1.00 | 0.89 | 1.00 | 1.00 |
| login-user-popup | 0.68 | 0.68 | 0.72 | 1.00 | 0.10 | 0.02 | n/a | 0.72 | 0.10 | 1.00 |
| login-user | 1.00 | 1.00 | 0.82 | 1.00 | 0.00 | 0.00 | 1.00 | 0.82 | 1.00 | 1.00 |
| moving-items | n/a | n/a | n/a | 0.88 | 0.69 | 0.13 | 0.78 | 0.13 | 0.69 | 0.88 |
| multi-layouts | 0.72 | 0.96 | 0.83 | 1.00 | 0.00 | 0.00 | 1.00 | 0.83 | 0.00 | 1.00 |
| multi-orderings | 1.00 | 1.00 | 0.88 | 1.00 | 0.00 | 0.00 | 1.00 | 0.88 | 0.00 | 1.00 |
| navigate-tree | 0.86 | 1.00 | 0.91 | 0.99 | 0.94 | 0.32 | 1.00 | 0.99 | 1.00 | 0.99 |
| number-checkboxes | n/a | n/a | n/a | 0.99 | 0.00 | 0.00 | 0.16 | 0.00 | 0.00 | 0.99 |
| read-table-2 | n/a | n/a | n/a | 0.94 | 0.00 | 0.00 | 0.00 | 0.00 | 0.00 | 0.94 |
| read-table | n/a | n/a | n/a | 0.97 | 0.00 | 0.01 | 0.00 | 0.01 | 0.00 | 0.97 |
| resize-textarea | n/a | n/a | n/a | 1.00 | 0.68 | 0.27 | 0.11 | 0.27 | 0.68 | 1.00 |
| right-angle | n/a | n/a | n/a | 0.98 | 0.98 | 0.26 | 0.38 | 0.26 | 0.98 | 0.98 |
| scroll-text-2 | n/a | n/a | n/a | 1.00 | 1.00 | 0.88 | 0.96 | 0.88 | 1.00 | 1.00 |
| scroll-text | n/a | n/a | n/a | 0.96 | 0.00 | 0.04 | 0.00 | 0.04 | 0.00 | 0.96 |
| search-engine | 1.00 | 1.00 | 0.34 | 1.00 | 0.01 | 0.15 | 1.00 | 0.34 | 1.00 | 1.00 |
| simon-says | n/a | n/a | n/a | 0.00 | 0.00 | 0.02 | 0.28 | 0.02 | 0.00 | 0.28 |

| | | | | | | | | | | |
|---|---|---|---|---|---|---|---|---|---|---|
| simple-algebra | 1.00 | 1.00 | n/a | 0.75 | 0.00 | 0.03 | 0.04 | 0.03 | 0.00 | 0.75 |
| simple-arithmetic | n/a | n/a | n/a | 0.86 | 0.00 | 0.38 | 0.07 | 0.38 | 0.00 | 0.86 |
| social-media-all | 1.00 | 1.00 | 0.00 | 0.75 | 0.00 | 0.00 | 1.00 | 0.00 | 1.00 | 0.75 |
| social-media-some | 0.90 | 0.96 | 0.02 | 0.85 | 0.02 | 0.01 | 0.42 | 0.02 | 0.02 | 0.85 |
| social-media | 0.98 | 0.98 | 0.21 | 0.90 | 0.02 | 0.03 | 1.00 | 0.24 | 1.00 | 0.90 |
| terminal | 1.00 | 1.00 | n/a | 0.00 | 0.00 | 0.00 | 0.00 | 0.00 | 0.00 | 0.00 |
| text-editor | n/a | n/a | n/a | 0.98 | 0.00 | 0.11 | 0.01 | 0.11 | 0.00 | 0.98 |
| text-transform | 0.80 | 0.80 | n/a | 0.60 | 0.10 | 0.19 | 0.00 | 0.19 | 0.10 | 0.60 |
| tic-tac-toe | 0.56 | 0.56 | 0.48 | 0.83 | 0.76 | 0.32 | 0.47 | 0.48 | 0.76 | 0.83 |
| unicode-test | n/a | n/a | n/a | 1.00 | 1.00 | 0.86 | n/a | 0.86 | 1.00 | 1.00 |
| use-autocomplete | 0.58 | 0.58 | 0.22 | 1.00 | 1.00 | 0.07 | 0.98 | 0.22 | 1.00 | 1.00 |
| use-colorwheel-2 | n/a | n/a | n/a | 0.95 | 0.85 | 0.38 | 1.00 | 0.38 | 0.85 | 1.00 |
| use-colorwheel | n/a | n/a | n/a | 0.98 | 0.82 | 0.68 | 1.00 | 0.68 | 0.82 | 1.00 |
| use-slider-2 | n/a | n/a | n/a | 0.95 | 0.00 | 0.03 | 0.15 | 0.03 | 0.00 | 0.95 |
| use-slider | n/a | n/a | n/a | 0.91 | 0.47 | 0.18 | 0.51 | 0.18 | 0.47 | 0.91 |
| use-spinner | 0.88 | 0.96 | 0.07 | 1.00 | 0.02 | 0.47 | 0.17 | 0.47 | 0.02 | 1.00 |
| visual-addition | n/a | n/a | n/a | 0.99 | 0.00 | 0.36 | 0.01 | 0.36 | 0.00 | 0.99 |

Table 18: Comprehensive task-level success rate evaluation of baseline models in MiniWoB++ tasks. *Ours (w/ GPT-4)* depicts the performance outcomes obtained through the use of the GPT-4 model for some tasks, which are visually highlighted in the color blue. The performance of baseline models has been sourced from prior studies [30, 24]. The average success rates of the tasks highlighted with violet color are shown in Figure 3. The state-of-the-art (SotA) in supervised learning (SL) is represented by the works of [30, 24] while the SotA in reinforcement learning (RL) includes the studies of [30, 25, 32]. Furthermore, the SotA in the combined application of SL and RL consists of the contributions of [30, 61, 36]. Combined result of models proposed prior to CC-Net [30] is denoted as *Others*, which include [61, 36, 25, 32]. This corresponds to *Aggregated SotA (Augmented)* baseline in previous works [30]. We generously estimate the performance of *CC-Net (RL)* based on their figures.

