# OpenReview forum: "Language Models can Solve Computer Tasks"
_NeurIPS.cc/2023/Conference — NeurIPS 2023 poster_

### Official Review · Reviewer_s5S9 · 2023-07-05

**Soundness:** 3 good
**Presentation:** 3 good
**Contribution:** 3 good
**Rating:** 6
**Confidence:** 4

**Summary:**

The paper introduces a novel approach, Recursively Criticizes and Improves (RCI), for guiding large language models (LLMs) to execute computer tasks using natural language commands. The RCI method outperforms existing methods and sets a new state-of-the-art on the MiniWoB++ benchmark when combined with the InstructGPT-3+RLHF LLM. The authors also demonstrate that RCI enhances LLMs' reasoning abilities on natural language reasoning tasks, and performs better when combined with the chain of thought (CoT) prompting.

**Strengths:**

1. The paper is overall well-written and easy to follow.
2. The paper introduces a new method, Recursively Criticizes and Improves (RCI), which guides large language models (LLMs) to execute computer tasks using natural language commands. The idea of explicit and implicit RCI is interesting.
3. Abundant experiments have shown that RCI not only shows great reasoning ability on different tasks, but also achieves same performance with fully-supervised methods on real-world computer tasks.

**Weaknesses:**

There are a lot of works that leverage environmental feedback or model self-reflection to improve the plans, like self-debugging, reflexion, self-refine, etc. The authors have briefly introduced some of them in the appendix, but a detailed comparison of these methods is still needed. The baselines included in the reasoning part of experiments are not enough as well.

**Questions:**

I am wondering how RCI performs compared with other LLMs-based methods that leverages environment feedback and self-evaluation.


**Limitations:**

The authors include the limitations in the model failure part.

---

> ### Author Rebuttal · Authors · 2023-08-09
>
> Thank you for your insightful feedback on our submission. We appreciate the time you took to review our work. We have carefully considered your comments and have made improvements to address the weaknesses and address your questions and concerns.
>
> 1. More comparison with other methods (e.g. self-debugging, reflexion, self-refine) that leverage environmental feedback
>
> Self-debugging, Reflexion, and Self-refine require external feedback from the environment to enhance their outputs. However, this implies that in the domain of decision-making, a reward function is needed. One of the main contributions of the RCI agent is removing the reliance on the reward function for various computer control tasks. Although the core concept of using self-feedback is similar, the problem we aim to solve using self-feedback is distinct from these methods. Thank you for pointing out that this distinction between our method and these related works was not clear. We will be sure to make this point clearer in the camera-ready version of the paper.
>
> 2. More baselines for reasoning tasks
>
> We disagree that the baselines included in our reasoning experiments are not enough. We use two baselines, CoT and Zero-Shot CoT which were the SoTA at the time of submission on the reasoning benchmarks we used. We carefully argue that it is sufficient to show the effectiveness of RCI by performing better than SoTA on a subset of reasoning benchmarks. Are you aware of another approach that outperformed CoT and Zero-Shot CoT on these benchmarks at the time of submission? If so we can happily include comparisons for the camera-ready version of the paper.

---

> > ### Comment · Reviewer_s5S9 · 2023-08-18
> > **Thanks the Authors' Responses**
> >
> > Thanks the authors for their responses. I will raise my score to 6, as most of my concerns have been resolved.

---

### Official Review · Reviewer_qKku · 2023-07-06

**Soundness:** 3 good
**Presentation:** 3 good
**Contribution:** 3 good
**Rating:** 6
**Confidence:** 4

**Summary:**

The paper presents a new method to enable LLMs to execute computer tasks guided by natural language. The method uses a simple prompting scheme called Recursive Criticism and Improvement (RCI), which prompts the LLMs to find and fix problems in their output. Without large amounts of expert demonstrations and task-specific reward functions, RCI improves the performance of LLMs on the MiniWoB++ benchmark, a web-based simulation environment that offers a diverse range of computer tasks. The paper also demonstrates that RCI enhances the reasoning abilities of LLMs on a suite of natural language reasoning tasks. The paper claims that RCI is a significant contribution to the development of intelligent agents that can solve any computer task by communicating via natural language.

**Strengths:**

1. This paper is original and insightful in tackling the grounding problem of LLMs by introducing RCI which uses the self-criticism and self-improvement abilities of LLMs to finish computer tasks and reasoning tasks.
2. It seems that RCI is general and flexible enough to be integrated with other techniques like CoT.
3. The paper applies RCI to a diverse range of web-based computer tasks and natural language reasoning tasks and shows promising results.
4. The paper is well-written and clear in presenting the main ideas, contributions, and results.

**Weaknesses:**

1. The impact of the RCI iteration steps should be presented. Will RCI iteration always increase the performance?
2. There is no explicit criterion for terminating RCI, and it relies on multiple attempts on the dataset. However, this kind of trial may not be feasible for new tasks in the real world. A simple approach might be to iterate as much as possible, but this would increase computational and time costs significantly.
3. Although RCI performs well in various reasoning tasks, considering the wide variety of models available now, it is recommended that the author conduct tests on several different models to demonstrate the effectiveness of RCI. For example, zero-shot CoT [1] has measured 17 different models in total. Another example is that few-hot CoT [2] has also measured models with different sizes (see their Table 4). Also, newly proposed models like Vicuna should be measured if possible.

[1] Large Language Models are Zero-Shot Reasoners

[2] Chain-of-Thought Prompting Elicits Reasoning in Large Language Models

**Questions:**

1. The model name “InstructGPT-3 + RLHF” in the experiment part of the main paper is quite confusing since there are 3 different models named after it in the appendix. Also, InstructGPT is also tuned with RLHF. I suggest replacing them with corresponding API names.
2. Why are benchmark AQuA, CommonSenseQA, and StrategyQA evaluated on the zero-shot setting but not evaluated on the few-shot setting?
3.“Specifically, we employ three models, namely, GPT-3 (davinci), InstructGPT-3 (text-davinci-002) , and InstructGPT-3 + RLHF (text-davinci-003, gpt-3.5-turbo, gpt-4).” Which InstructGPT-3 + RLHF model is used in section 3.2?
4. Line 246 – 252: “Our findings reveal that our agent surpasses SotA in supervised learning by 34% and in reinforcement learning by 24%. Notably, our proposed RCI prompting method outperforms the SotA LLM approach, even when the latter employs both finetuning and few-shot examples in prompts. This outcome highlights the effectiveness of our approach in extracting vital knowledge for computer tasks from language models. However, our agent underperforms in comparison to SotA in the combination of supervised and reinforcement learning, which employs significantly more expert data and online interactions.” This result is inconsistent with Figure 4 (a). I guess there is a typo in Figure 4 (a) where “Ours” should be 0.91 instead of 0.94.

**Limitations:**

The authors have not discussed their limitations.

---

> ### Author Rebuttal · Authors · 2023-08-09
>
> Thank you for the review and valuable feedback. We appreciate the reviewer's careful examination of our submission. Below is our rebuttal addressing the raised concerns:
>
> 1. Impact of RCI iteration steps
> We agree that the impact of RCI iteration steps should have been explicitly presented in the paper. While it is true that RCI iteration does not always increase performance because of false negative critics, we found that if we properly set the maximum iteration number, it didn’t cause a notable performance drop. We also acknowledge that the lack of an explicit criterion for terminating RCI is not well specified. In our evaluation, we could find that empirically stopping the interaction after 2 showed the best performance, but more heuristics and advanced methods can further improve this process.
>
> 2. Evaluation on multiple models
> Although we didn’t have enough budget to run on multiple models for this submission, this is an excellent suggestion, and we will try to include additional models for the camera-ready version.
>
> 3. Confusing model naming
> We apologize for the confusion caused by the naming of the models in the experiment section. In the revised paper, we will make clear which API is used for each section of the evaluations to avoid any ambiguity.
>
> 4. Benchmarks for few-shot setting
> The reason we didn’t evaluate AQuA, CommonSenseQA, and StrategyQA on the few-shot setting was that the previous works on the few-shot setting were not evaluated on the tasks. Since the few-shot approach highly relies on the few-shot examples, it was hard to set up a fair experiment setting without those data.
>
> 5. InstructGPT-3 + RLHF model used in section 3.2
> We apologize for the oversight. In section 3.2, we used the "gpt-3.5-turbo" variant of the InstructGPT-3 + RLHF model for the experiments. We will update the paper to clarify this.
>
> 6. Inconsistency between text and Figure 4(a)
> We thank the reviewer for properly pointing out the mismatch between the text and figure 4(a). There seems to be an obsolete explanation for Figure 4(a) regarding the performance comparison. We apologize for this mistake. In the revised version, we will correct the text to be aligned with the figure.
>
> 7. Limitations:
> Thank you for bringing this to our attention. We acknowledge that our initial discussion of limitations may have been inadequate, and we appreciate the opportunity to address this concern in our revised version. In the revised version, we will address the following main limitations of our work:
>
> * Limited Context Length: Due to the maximum context length of LLMs, lengthy HTML states hinder the use of LLMs for web navigation.
> * Limited Actions Space: Currently, our model only supports click and keyboard typing actions.
> * Lack of Long-Term Strategies: Our model focuses on short-term decisions and lacks long-term strategic planning capabilities.
>
> Once again, we appreciate the reviewer's thorough evaluation and constructive feedback. We will address all the concerns raised in the paper's revised version to strengthen the quality and clarity of our work.

---

> > ### Comment · Reviewer_qKku · 2023-08-16
> >
> > Thanks for the author's rebuttal! Some of my concerns have been addressed.
> >
> > My remaining concern is that the authors have not provided the evaluation of multiple models to show the method's generalization ability (weakness 3) so I keep my score at 6-weak accept for now.

---

### Official Review · Reviewer_6D7X · 2023-07-07

**Soundness:** 3 good
**Presentation:** 3 good
**Contribution:** 3 good
**Rating:** 6
**Confidence:** 5

**Summary:**

This paper proposes a prompting approach that utilizes a critic to improve the web navigation performance. Within the same context, the model reviews a candidate answer and improves it based on reasoning steps. For web navigation, there are two phases: explicit and implicit RCI. Explicit RCI takes a task and generates a plan with step-by-step actions. This plan is fed to the implicit RCI to ground on the state and the agent. Resulting action is executed in the environment to retrieve a new state. Compared to previous SL and RL approach, RCI achieves competitive to the SoTA while using only few-shot prompting.

**Strengths:**

The paper introduces a novel prompting approach with planning and iterative improvement. It performs favorably to CoT prompting and outperforms previous finetuning approaches while using significantly less data. It also shows the general capabilities of LLMs in decision making tasks and illustrates limits of the MiniWoB benchmark.

**Weaknesses:**

Some questions regarding better baselines and RLHF performance.

- More detailed comparison with recent approaches such as ReACT or InnerMonologue is missing. What is the main novelty of RCI and why would it be a better candidate for computer tasks compared to ReACT which also studies WebShop benchmark tasks.

- Instruction tuning and RLHF are crucial for achieving good results, otherwise the success rate drops to 50%. Given that training data for RLHF training is undisclosed, I wonder if similar tasks (or even MiniWoB data) are used during training. Is there any empirical evidence for the case of InstructGPT-3+RLHF model?

- WebN-T5 doesn't use in-context examples. It uses the history of actions within the same episode. (referring to Figure-4 in your paper)

- WebGUM and WebN-T5 also don't use dictionary based typing actions. WebGUM is missing from comparison.

- Please clarify Line 33 as there are previous models using LLMs that you also compare.

**Questions:**

1. How do you compare to more recent reasoning approaches?

2. Could RLHF training be using some web navigation tasks or even MiniWoB?

**Limitations:**

The paper discusses some of the limitations but more detailed discussion on generalization to real websites would be helpful.

---

> ### Author Rebuttal · Authors · 2023-08-09
>
> We express our gratitude to the reviewer for their valuable feedback, which has significantly contributed to identifying areas for improvement in our paper. Below, we address the points raised to enhance the clarity of our content.
>
> 1. Comparison with ReACT and InnerMonologue on computer tasks
> Regarding the comparison with ReACT and InnerMonologue on computer tasks, it is important to note that Webshop has a more limited action space, defining a single task, which only involves shopping on the internet. We consider it analogous to the tasks implemented in the MiniWoB++ benchmark.
> Direct feedback from LLMs, which includes information about invalid actions that cannot be executed by the agent or is invalid in the current state, proves to be highly beneficial in the RCI process. Critics, with their explicit explanation, offer valuable guidance for improvement, making the RCI approach advantageous compared to mere reasoning without any clear direction for enhancement. While ReACT can be integrated into the RCI process, we firmly believe that critics play a crucial role in grounding actions for the state, task, and agent, leading to more effective outcomes.
> Reflexion, a more similar work to ours, builds upon ReACT and employs reasoning steps to enhance task and state grounding of action generations. However, it relies on external feedback, indicating a continued need for a reward function. In contrast, the RCI agent does not depend on any external reward function.
> We would also like to note that a more comprehensive comparison with related works, including ReACT and InnerMonologue, has been provided in the related works section of the appendix.
>
>
> 2. Empirical evidence for InstructGPT-3+RLHF model training data
> The reviewer inquired about empirical evidence that the InstructGPT-3+RLHF model is not trained with similar tasks. While we do not have specific evidence regarding the exact training data, we believe it is unlikely that expert demonstration data from MiniWoB++ was used for the training of InstructGPT-3+RLHF. The use of HTML code for training is likely, but we believe that extensive demonstration data specific to MiniWoB++ tasks is improbable because to our knowledge the RLHF was done in the context of a chatbot, not a computer agent. We do not know of experiments that can be run to determine if InstructGPT-3+RLHF used MiniWoB demonstrations, but are open to suggestions. Unfortunately, we will not definitively know the answer and we acknowledge the downsides of doing academic research on closed-source LLMs. In our revised manuscript we will include discussion on this topic.
>
>
> 3. Clarification
> We appreciate the reviewer pointing out the confusion in Line 33 regarding the best-performing approach. Indeed, CC-Net, which does not use LLMs, was the best-performing model in our comparisons. We apologize for the oversight and will make the necessary corrections in the revised version. Furthermore, we will accurately represent the usage of dictionary-based actions in the models for a more coherent comparison.
>
> 4. Comparison with WebGUM
> As the WebGUM paper was published after the deadline of our paper, we weren’t able to include it. We will cite this paper as contemporaneous work in the camera-ready version, but we note that WebGUM performs worse than our approach.
>
> 5. Comparison to more recent reasoning approaches
> In the camera-ready version of the paper we can include comparisons with newer reasoning approaches such as Tree of Thoughts and Self-Notes, but since these works came out after we submitted the paper we do not include these comparisons in the current version.
>
> 6. Does RLHF training use web navigation?
> As discussed above, we will include qualifications and limitations that we do not know for certain how these models are trained, which is a major drawback of doing academic research on closed-source LLMs.
>
> 7. Limitations: Generalization to real websites
> The reviewer also raised a concern regarding the generalization of this work to real websites. We acknowledge that there are certain limitations, such as the restricted context length and limited action space, which may hinder the direct application of the RCI agent to real websites. Additionally, the presence of complex UI components that cannot be fully understood by HTML code might also impede further generalization. Nevertheless, it is important to note that our approach is intended to be viewed as complementary and independent from these orthogonal challenges that are left for future work.

---

> > ### Comment · Reviewer_6D7X · 2023-08-21
> > **Thanks for your response!**
> >
> > I think solving computer tasks using a strong prompting approach is valuable, so I will maintain my score. But, I am still skeptical of the proposed approach's generalizability to other models other than InstructGPT-3+RLHF. Related to that, the question of whether any web navigation related task is utilized in RLHF or instruction tuning makes the contribution of the proposed approach less clear.

---

### Official Review · Reviewer_kDG8 · 2023-07-07

**Soundness:** 3 good
**Presentation:** 3 good
**Contribution:** 3 good
**Rating:** 6
**Confidence:** 4

**Summary:**

This paper proposes an algorithm for executing tasks using LLMs. While this can in principle be applied to any sequential task with language input and output spaces, they tested primarily on computer control tasks.

The method, RCI, works as follows:
- The environment state, action space, and task are represented in language (e.g. HTML code for the state).
- Based on the task description, the LLM generates a high-level plan of how to solve the task.
- The LLM writes a critique of its plan and improves it. This can happen multiple times.
- Conditioned on the plan and the current state, the LLM is asked to find the webpage elements necessary to execute the plan.
- Based on the above, ask the LLM to generate an executable action.

They show that on a large subset of Miniwob++ tasks, this method is able to outperform past LLM SOTA and is able to match past IL+RL SOTA without needing any training. They also show improvements over other prompting methods on various math and reasoning tasks.

**Strengths:**

- The method gives a substantial performance boost compared to all Miniwob++ baselines (except one which uses far more demonstration data). It also shows consistent improvements in math and reasoning tasks. This is useful since it suggests that for many computer control tasks collecting lots of demo or RL data is unnecessary.
- Each component of the pipeline includes intuition for why it's necessary + ablations showing that performance drops significantly without it.
- It's hard to judge novelty given the recent explosion in works prompting LLMs for tasks with various forms of self-reflection. I'm inclined to say it's all concurrent work.

**Weaknesses:**

- See the "Questions" section for discussion on figure/discussion mismatch.
- This method has limitations, including (a) it is hard to solve tasks where the HTML description exceeds the context width and (b) the agent is currently unable to handle tasks which involve dragging or other complex UI manipulation. Currently, these tasks are excluded from the set of evaluation tasks. On the other hand, tasks where the prior SotA did poorly were stated to be intentionally included. This seems like a bit of an unfair comparison.  (Though I think many of these problems could be addressed by modifying the action space and getting a longer-context LM). Even if the additional tasks are not reported in the main results I think it should at least be more prominently emphasized that tasks where RCI has insufficient context were modified or excluded.

**Questions:**

**Major questions**
* The description of Figure 4a doesn't seem to quite match the numbers in the table. "our agent surpasses SotA in supervised learning by 34% and in reinforcement learning by 24%".  (I'm guessing you meant 37 and 27. Also, please be clear whether you're talking about *percent improvement* or *percentage point improvement.*). Also, the text says "our agent underperforms in comparison to SotA in the combination of supervised and reinforcement learning..." but the graph currently shows they match. Which is correct?

**More minor**
* Nit: Typo L196 "wich"
* Nit: L322 "exist" -> "exists"
* Explain more clearly what "dictionary-based typing actions" are.
* In Figure 4a, it would be helpful if the caption stated which models are the "SL SotA", "RL SotA", and "SL + RL SotA". Currently it isn't clear.


**Limitations:**

Limitations are discussed, although additional limitations (such as lack of history, limited context window, and inability to improve through accumulated agent experience) are not discussed much.

---

> ### Author Rebuttal · Authors · 2023-08-09
>
> We appreciate the reviewers' valuable feedback on our paper. We have carefully considered the comments and suggestions and would like to address each of the concerns raised.
>
> 1. Task selection:
> We understand the concern about task selection for our evaluation. We followed the WebN-T5-3B's [1] task set selection as it was the most recent work at the time and the first to use LLMs for solving computer tasks. We acknowledge that this approach might have resulted in a limited comparison. To address this, we will explicitly mention in the paper that the task set was selected to provide a direct comparison with the most relevant previous work at the time. We also acknowledge the importance of considering other relevant tasks and will make this point clear in the revised manuscript.
>
> 2. Context length limitation for lengthy HTML state:
> We agree that HTML descriptions exceeding the context length pose a challenge. While we used the original HTML code, we recognize the possibility of reducing the HTML state size by removing irrelevant code, such as CSS and JS code. This optimization could improve the model's ability to handle tasks with lengthy HTML effectively. We will incorporate this suggestion into the discussion section of the revised paper.
>
> 3. Limited action space:
> The reviewer correctly pointed out that the action space definition is limited. We acknowledge the importance of a more extensive action space to support diverse computer tasks. However, as the reviewer already mentioned, we expect that it can be achieved by adding more feasible actions to the prompt. This modification should enhance the model's capability to handle tasks involving dragging or other complex UI manipulations.
>
> 4. Mismatch between figures and explanation:
> We apologize for the oversight in the description of Figure 4a. The discrepancy in the numbers appeared in the figure and the text will be rectified in the revised version.
>
> Furthermore, we will address the minor nitpicks mentioned, including the typos in Lines 196 and 322, as well as provide a clearer explanation of "dictionary-based typing actions."
>
> Lastly, we appreciate the reviewer pointing out the limitations not fully discussed in the paper. Our agent keeps the history of actions in the prompt, but we will make sure that other limitations such as limited context window, and inability to improve through accumulated agent experience will be properly discussed.
>
> In conclusion, we thank the reviewers for their insightful comments, which have helped us improve the quality of our work.
>
> [1] UNDERSTANDING HTML WITH LARGE LANGUAGE MODELS

---

> > ### Comment · Reviewer_kDG8 · 2023-08-16
> > **Thanks**
> >
> > Thanks for answering my questions. I will maintain my score.

---

### Decision · Program_Chairs · 2023-09-21

**Decision:**

Accept (poster)

**Comment:**

This work proposes using a simple prompting scheme where an LLM agent recursively criticizes and improves its output (RCI). It is shown that RCI outperforms SL and RL on the MiniWoB++ benchmark. It is interesting and impressive to see that this best performing method requires no training.

The authors addressed most of the concerns raised by the reviewers. After the rebuttal, the reviewers unanimously recommended (weak) accept.

Reviewers recognize the performance of the proposed method as well as the clear presentations and intuitions.

The only remaining concern is regarding the generalization ability of the model to other tasks (6D7X, qKku). We think that without detailed analysis on generalization ability, the work is still of great interest to the community. That said, please try to provide more discussions in the final version.